# Runx2 is essential for the transdifferentiation of chondrocytes into osteoblasts

Xin Qin[1]☯, Qing Jiang[1]☯, Kenichi Nagano[2], Takeshi Moriishi[3], Toshihiro Miyazaki[3], Hisato Komori[1], Kosei Ito[4], Klaus von der Mark[5], Chiharu Sakane[6], Hitomi Kaneko[1], Toshihisa Komori[1]*

1 Basic and Translational Research Center for Hard Tissue Disease, Nagasaki University Graduate School of Biomedical Sciences, Nagasaki, Japan, 2 Department of Oral Pathology and Bone Metabolism, Nagasaki University Graduate School of Biomedical Sciences, Nagasaki, Japan, 3 Department of Cell Biology, Nagasaki University Graduate School of Biomedical Sciences, Nagasaki, Japan, 4 Department of Molecular Bone Biology, Nagasaki University Graduate School of Biomedical Sciences, Nagasaki, Japan, 5 Department of Experimental Medicine I, Nikolaus-Fiebiger Center of Molecular Medicine, Friedrich-Alexander University Erlangen-Nürnberg, Erlangen, Germany, 6 Division of Comparative Medicine, Life Science Support Center, Nagasaki University, Nagasaki, Japan

☯ These authors contributed equally to this work.
* komorit@nagasaki-u.ac.jp

Data Availability Statement: All relevant data are within the manuscript and its Supporting Information files.

## Abstract

Chondrocytes proliferate and mature into hypertrophic chondrocytes. Vascular invasion into the cartilage occurs in the terminal hypertrophic chondrocyte layer, and terminal hypertrophic chondrocytes die by apoptosis or transdifferentiate into osteoblasts. Runx2 is essential for osteoblast differentiation and chondrocyte maturation. *Runx2*-deficient mice are composed of cartilaginous skeletons and lack the vascular invasion into the cartilage. However, the requirement of Runx2 in the vascular invasion into the cartilage, mechanism of chondrocyte transdifferentiation to osteoblasts, and its significance in bone development remain to be elucidated. To investigate these points, we generated *Runx2*fl/flCre mice, in which Runx2 was deleted in hypertrophic chondrocytes using *Col10a1* Cre. Vascular invasion into the cartilage was similarly observed in *Runx2*fl/fl and *Runx2*fl/flCre mice. Vegfa expression was reduced in the terminal hypertrophic chondrocytes in *Runx2*fl/flCre mice, but Vegfa was strongly expressed in osteoblasts in the bone collar, suggesting that Vegfa expression in bone collar osteoblasts is sufficient for vascular invasion into the cartilage. The apoptosis of terminal hypertrophic chondrocytes was increased and their transdifferentiation was interrupted in *Runx2*fl/flCre mice, leading to lack of primary spongiosa and osteoblasts in the region at E16.5. The osteoblasts appeared in this region at E17.5 in the absence of transdifferentiation, and the number of osteoblasts and the formation of primary spongiosa, but not secondary spongiosa, reached to levels similar those in *Runx2*fl/fl mice at birth. The bone structure and volume and all bone histomophometric parameters were similar between *Runx2*fl/fl and *Runx2*fl/flCre mice after 6 weeks of age. These findings indicate that Runx2 expression in terminal hypertrophic chondrocytes is not required for vascular invasion into the cartilage, but is for their survival and transdifferentiation into osteoblasts, and that the transdifferentiation is necessary for trabecular bone formation in embryonic and neonatal stages, but not for acquiring normal bone structure and volume in young and adult mice.

**Funding:** This work was supported by grants from the Japanese Ministry of Education, Culture, Sports, Science and Technology (https://www.jsps.go.jp/) to TK (Grant number: 18H05283), XQ (19K24124), and QJ (20K18460). The funders had no role in study design, data collection and analysis, decision to publish, or preparation of the manuscript.

**Competing interests:** The authors have declared that no competing interests exist.

## Author summary

Chondrocytes and osteoblasts are different lineage cells, which differentiate from mesenchymal stem cells through the regulation by different transcription factors. In endochondral bone development, chondrocytes proliferate and mature to terminally differentiated chondrocytes, and vascular invasion occurs in the layer of terminally differentiated chondrocytes. Terminally differentiated chondrocytes die by apoptosis or transdifferentiate into osteoblasts, which contribute to bone formation. However, the molecular mechanism and physiological significance of the transdifferentiation remain to be clarified. Runx2 is an essential transcription factor for osteoblast differentiation and chondrocyte maturation and has been considered to be required for vascular invasion into the cartilage. By deleting Runx2 in differentiated chondrocytes, we elucidated that Runx2 is essential for the transdifferentiation and is required for maintaining the survival of terminally differentiated chondrocytes but not for vascular invasion into the cartilage. Furthermore, we clarified that the transdifferentiation is required for trabecular bone formation in embryonic and neonatal stages, but that it is dispensable for acquiring normal bone structure and volume in young and adult mice probably due to the major contribution of osteoblasts originated from perichondrium/periosteum.

## Introduction

Vertebral skeletons are formed through intramembranous ossification or endochondral ossification. In intramembranous ossification, bone is directly formed by osteoblasts. In endochondral ossification, mesenchymal cells condense and differentiate into chondrocytes, and cartilaginous skeletons are first formed by chondrocytes, which express *Col2a1*. Chondrocytes proliferate and differentiate (maturate) to prehypertrophic chondrocytes, which express *Ihh* and *Pth1r*, and then to hypertrophic chondrocytes, which express *Col10a1*. The hypertrophic chondrocytes further differentiate (mature) into terminal hypertrophic chondrocytes, which express *Mmp13*, *Ibsp*, and *Spp1*, the matrix is mineralized, blood vessels and osteoblast lineage cells in the perichondrium invade the calcified cartilage, the matrix of cartilage is resorbed by osteoclasts, and the cartilage is replaced with bone. During chondrocyte maturation, perichondrial cells differentiate into osteoblasts and form the bone collar through intramembranous ossification [1, 2].

Runx2 is a transcription factor that belongs to Runx family, which is composed of Runx1, Runx2, and Runx3 [3]. Runx2 is expressed in osteoblast progenitors and osteoblasts, and is essential for the proliferation of osteoblast progenitors and their differentiation into osteoblasts [4]. Runx2 is also expressed in chondrocytes, and its expression is upregulated in prehypertrophic chondrocytes and is maintained until terminal hypertrophic chondrocytes [2, 5]. *Runx2*-deficient (*Runx2*$^{-/-}$) mice are composed of cartilaginous skeletons [6, 7]. Chondrocyte maturation is inhibited in *Runx2*$^{-/-}$ mice and hypertrophic chondrocytes are absent in most of the skeleton except the tibia, fibula, radius, and ulna, in which chondrocytes have fully matured into terminal hypertrophic chondrocytes [2, 5]. However, vascular invasion into the cartilage is absent in whole skeletons of *Runx2*$^{-/-}$ mice, probably due to the reduced Vegfa expression in the terminal hypertrophic chondrocytes [8, 9]. Runx2 and Runx3 are essential for chondrocyte maturation and Runx2 plays a major role in this process [10–13]. Runx2 upregulates Ihh expression in prehypertrophic chondrocytes, and Ihh regulates chondrocyte proliferation directly and regulates chondrocyte maturation through Pthlh [10, 14, 15]. Ihh is also

required for Runx2 expression in perichondrial cells and their differentiation into osteoblasts [15].

In the process of endochondral ossification, the terminal hypertrophic chondrocytes die by apoptosis or transdifferentiate into osteoblasts [16–19]. Furthermore, it was recently reported that the deletion of *Ctnnb1* in hypertrophic chondrocytes increases bone resorption by increasing the Rankl/Opg ratio and reduces the number of transdifferentiated osteoblasts from chondrocytes, leading to a reduction in trabecular bone during the embryonic stage [20]. However, the molecular mechanism of the transdifferentiation of chondrocytes to osteoblasts is poorly understood and its role in bone development remain to be clarified. Moreover, the requirement of Runx2 expression in terminal hypertrophic chondrocytes for vascular invasion into the cartilage needs to be investigated by hypertrophic chondrocyte-specific deletion of *Runx2*. In this study, we generated *Runx2* conditional knockout (*Runx2*$^{fl/flCre}$) mice, in which *Runx2* was deleted in hypertrophic chondrocytes using *Col10a1* promoter-Cre. We demonstrated that Runx2 expression in terminal hypertrophic chondrocytes is essential for their transdifferentiation to osteoblasts, but not for vascular invasion into the cartilage, and that the transdifferentiation is transiently required for trabecular bone formation, but dispensable to acquire normal bone mass in young and adult mice.

## Results

### Skeletal development was similar in *Runx2*$^{fl/fl}$ and *Runx2*$^{fl/flCre}$ mice at E15.5, although apoptosis of terminal hypertrophic chondrocytes increased in *Runx2*$^{fl/flCre}$ mice

*Runx2*$^{fl/fl}$ mice were generated as described in Materials and Methods and shown in S1 Fig. *Runx2*$^{fl/fl}$ mice were mated with *Col10a1*-Cre transgenic mice [21] to generate *Runx2*$^{fl/flCre}$ mice, in which *Runx2* was specifically deleted in hypertrophic chondrocytes. On immunohistochemical analysis using anti-Runx2 antibody, the number of Runx2-positive hypertrophic chondrocytes in tibiae of *Runx2*$^{fl/flCre}$ mice was less than one twenty-fifth of that in wild-type mice at E15.5 (Fig 1A–1C). Skeletal preparation revealed that mineralization was similar in *Runx2*$^{fl/fl}$ and *Runx2*$^{fl/flCre}$ mice at E15.5 (Fig 1D and 1E). On histological analysis of femurs, hematoxylin and eosin (H-E), von Kossa, alkaline phosphatase (ALP), and safranin O staining confirmed the presence of chondrocyte hypertrophy, their terminal differentiation, vascular invasion into the cartilage, and bone collar formation in both *Runx2*$^{fl/fl}$ and *Runx2*$^{fl/flCre}$ mice at similar levels (Fig 1F–1M). However, the apoptosis of terminal hypertrophic chondrocytes was increased in *Runx2*$^{fl/flCre}$ mice compared with that in *Runx2*$^{fl/fl}$ mice (Fig 1N–1P). We compared the expression of the genes in hypertrophic and terminal hypertrophic layers in femurs and tibiae between *Runx2*$^{fl/fl}$ and *Runx2*$^{fl/flCre}$ mice at E15.5 using laser capture microdissection system (Fig 1Q). The Runx2 expression in *Runx2*$^{fl/flCre}$ mice was severely reduced and the *Col10a1* expression was similar to that in *Runx2*$^{fl/fl}$ mice. The expression of *Mmp13*, *Ibsp*, *Spp1*, and *Vegfa*, which are expressed in terminal hypertrophic chondrocyte layer, in *Runx2*$^{fl/flCre}$ mice was less than that in *Runx2*$^{fl/fl}$ mice (Fig 1R). Further, the expression of *Bcl2*, *Pten*, *Fas*, and *Bnip3* in *Runx2*$^{fl/flCre}$ mice was higher than that in *Runx2*$^{fl/fl}$ mice, suggesting that Pten, Fas, and Bnip3 may be involved in the increased apoptosis in *Runx2*$^{fl/flCre}$ mice (Fig 1R). These findings indicate that the deletion of *Runx2* in the hypertrophic chondrocytes did not affect the terminal differentiation of chondrocytes, vascular invasion into the calcified cartilage, bone collar formation, and the expression of *Col10a1*, but decreased the expression of *Mmp13*, *Ibsp*, *Spp1*, and *Vegfa* in the terminal hypertrophic chondrocytes and increased the apoptosis of terminal hypertrophic chondrocytes.

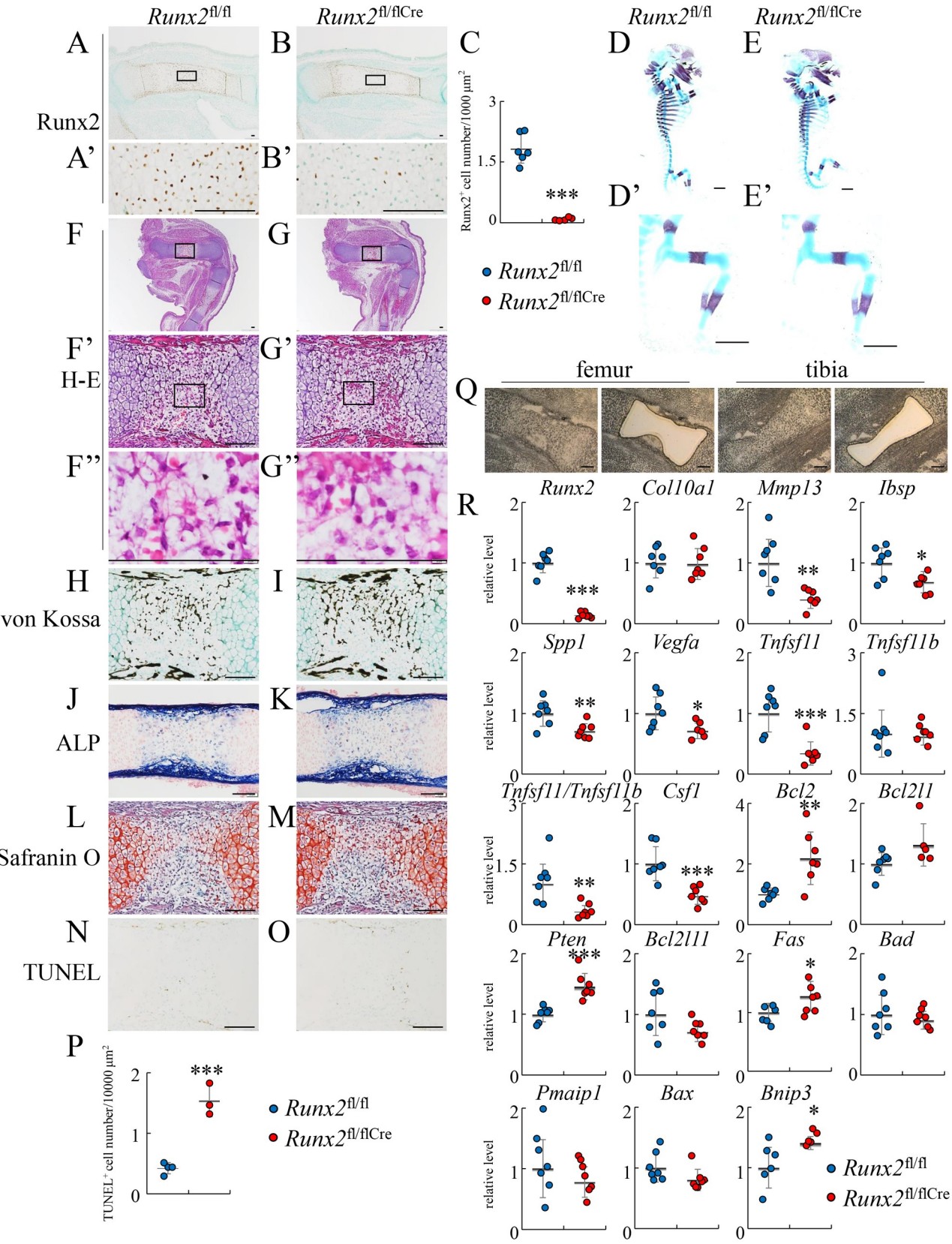

**Fig 1. Skeletal system and histological analyses in *Runx2*<sup>fl/fl</sup> and *Runx2*<sup>fl/fl/Cre</sup> embryos at E15.5.** (A and B) Immunohistochemical analysis in tibia from *Runx2*<sup>fl/fl</sup> (A) and *Runx2*<sup>fl/fl/Cre</sup> (B) mice using anti-Runx2 antibody. The boxed regions in A and B are magnified in A' and B', respectively. (C) The number of Runx2-positive hypertrophic chondrocytes was counted and shown. (D and E) Lateral view of the whole skeletons of *Runx2*<sup>fl/fl</sup> (D) and *Runx2*<sup>fl/fl/Cre</sup> (E) mice. Hind limbs were magnified in D' and E'. (F-P) Histological analyses using femoral sections from *Runx2*<sup>fl/fl</sup> (F, H, J, L, N) and *Runx2*<sup>fl/fl/Cre</sup> (G, I, K, M, O) mice. (F and G) H-E staining. The boxed regions in F, G, F', and G' are magnified in F', G', F", and G", respectively. (H and I) von Kossa staining. (J and K) ALP staining. (L and M) Safranin O staining. (N and O) TUNEL staining. (P) The number of TUNEL-positive hypertrophic chondrocytes was counted. Scale bars: 100 μm (A-B', F-O), 1 mm (D-E'). The number of mice analyzed: *Runx2*<sup>fl/fl</sup>: 4–6, *Runx2*<sup>fl/fl/Cre</sup>: 3–5. (Q and R) The expression of the genes related to chondrocyte and osteoclast differentiation and apoptosis in hypertrophic and terminal hypertrophic layers. (Q) The hypertrophic and terminal hypertrophic layers in femurs and tibiae at E15.5 were isolated using laser capture microdissection system and RNA was prepared. (R) Real-time RT-PCR analysis. The values in *Runx2*<sup>fl/fl</sup> mice were defined as 1, and relative levels are shown. The number of mice analyzed: *Runx2*<sup>fl/fl</sup>: 7, *Runx2*<sup>fl/fl/Cre</sup>: 7. Data are the mean ± SD. *Versus *Runx2*<sup>fl/fl</sup>, *p<0.05, **p<0.01, ***p<0.001.

## The number of osteoblasts in the bone marrow was markedly reduced and the formation of primary spongiosa was impaired in *Runx2*<sup>fl/flCre</sup> mice at E16.5

As skeletal development was histologically indistinguishable between *Runx2*<sup>fl/fl</sup> and *Runx2*<sup>fl/flCre</sup> mice at E15.5 when vascular invasion into the cartilage starts, the expression of chondrocyte and osteoblast marker genes was examined by in situ hybridization using femurs at E16.5 when the primary spongiosa is formed. On H-E staining, the primary spongiosa was formed in *Runx2*<sup>fl/fl</sup> and *Runx2*<sup>fl/+Cre</sup> mice but not in *Runx2*<sup>fl/flCre</sup> mice (Fig 2A–2C). The expression of *Col2a1* and *Col10a1* was similar among *Runx2*<sup>fl/fl</sup>, *Runx2*<sup>fl/+Cre</sup>, and *Runx2*<sup>fl/flCre</sup> mice (Fig 2D–2I and 2V). The expression of *Mmp13*, which is expressed in terminal hypertrophic chondrocytes, was reduced in *Runx2*<sup>fl/flCre</sup> mice as compared with *Runx2*<sup>fl/fl</sup> and *Runx2*<sup>fl/+Cre</sup> mice (Fig 2J–2L and 2V). The expression of *Ibsp* and *Spp1* was observed in terminal hypertrophic chondrocytes and osteoblasts in the bone collar and primary spongiosa in *Runx2*<sup>fl/fl</sup> and *Runx2*<sup>fl/+Cre</sup> mice, whereas it was observed in terminal hypertrophic chondrocytes and osteoblasts in the bone collar but not in the bone marrow in *Runx2*<sup>fl/flCre</sup> mice (Fig 2M–2R and 2V). The expression of *Col1a1*, which is expressed in osteoblasts, was strongly detected in osteoblasts in the bone collar and primary spongiosa in *Runx2*<sup>fl/fl</sup> and *Runx2*<sup>fl/+Cre</sup> mice, whereas it was strongly detected in osteoblasts in the bone collar but almost absent in bone marrow in *Runx2*<sup>fl/flCre</sup> mice (Fig 2S–2V). ALP staining revealed that ALP-positive osteoblasts had accumulated in the bone collar and bone marrow in *Runx2*<sup>fl/fl</sup> mice, and that they were also abundant in the bone collar but almost absent in the middle of the bone marrow in *Runx2*<sup>fl/flCre</sup> mice (Fig 3A and 3B). Runx2-positive osteoblasts and Sp7-positive osteoblasts were abundant in the bone collar and primary spongiosa in femurs of *Runx2*<sup>fl/fl</sup> mice, whereas they were abundant in the bone collar but markedly reduced in bone marrow in *Runx2*<sup>fl/flCre</sup> mice (Fig 3C–3H). The staining of endothelial cells with anti-CD34 antibody indicated that blood vessels invaded into the cartilage at a similar level in *Runx2*<sup>fl/fl</sup> and *Runx2*<sup>fl/flCre</sup> mice (Fig 3I–3K). The number of BrdU-positive cells was similar in proliferating layers of the growth plates in femurs between *Runx2*<sup>fl/fl</sup> and *Runx2*<sup>fl/flCre</sup> mice, whereas that in the region of presumptive primary spongiosa in *Runx2*<sup>fl/flCre</sup> mice was less than that in *Runx2*<sup>fl/fl</sup> mice (Fig 3L–3O). These findings indicate that the deletion of Runx2 in hypertrophic chondrocytes resulted in impaired formation of the primary spongiosa due to the marked reduction of osteoblasts in bone marrow irrespective of the normal vascular invasion into the cartilage.

Based on safranin O staining of femoral sections, there was more cartilage matrix in bone marrow in *Runx2*<sup>fl/flCre</sup> mice than in *Runx2*<sup>fl/fl</sup> mice, although the number of TRAP-positive osteoclasts in *Runx2*<sup>fl/flCre</sup> mice was higher than that in *Runx2*<sup>fl/fl</sup> mice (Fig 3P–3S, 3V and 3W). The number of TUNEL-positive cells was increased in the safranin O-stained region in the bone marrow of *Runx2*<sup>fl/flCre</sup> mice compared with *Runx2*<sup>fl/fl</sup> mice (Fig 3P', 3Q', 3T, 3U and 3X), indicating again that the deletion of *Runx2* in hypertrophic chondrocytes induces apoptosis in terminal hypertrophic chondrocytes. We compared the expression of *Tnfsf11*, *Tnfrsf11b*, and *Csf1* using

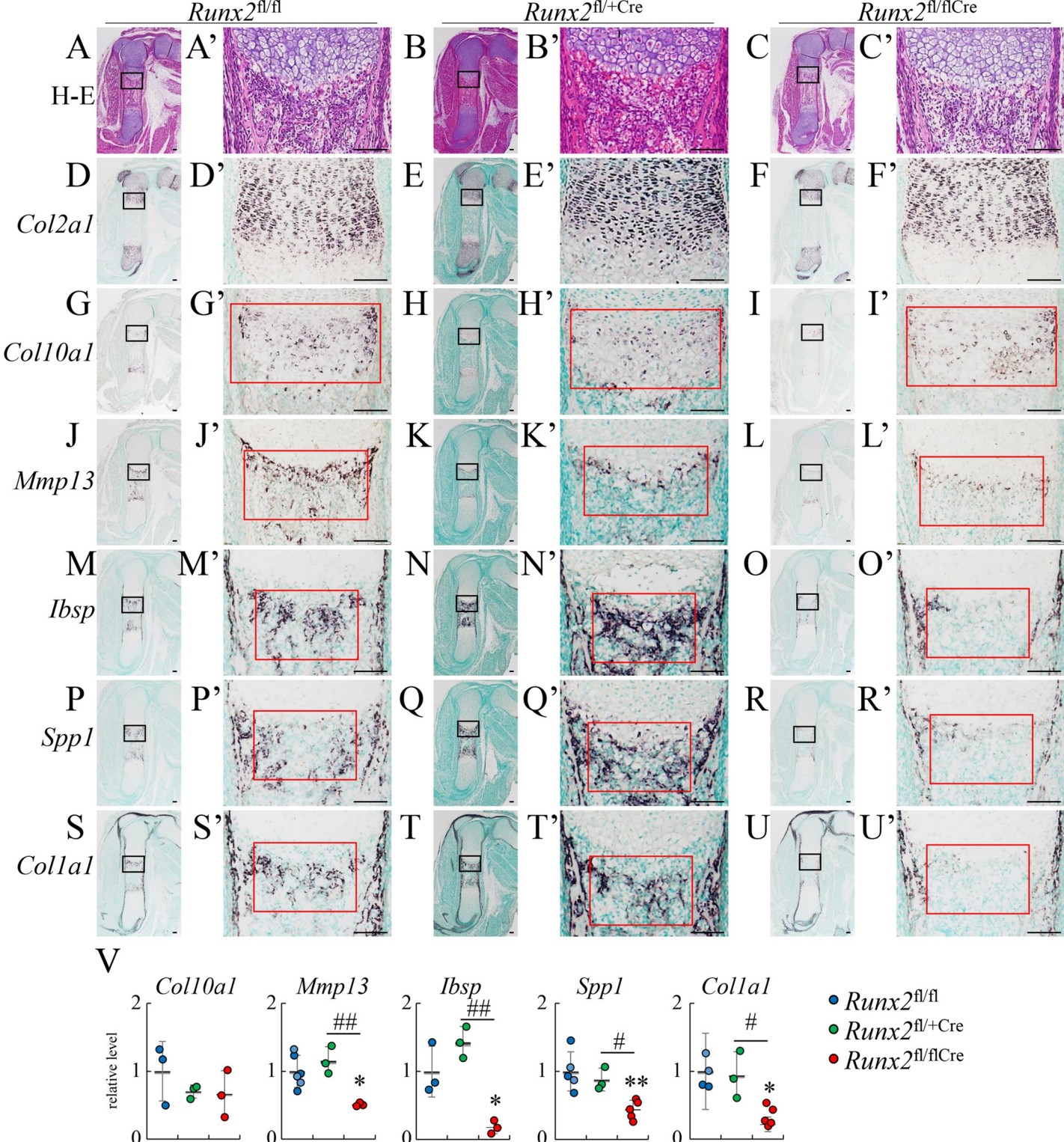

**Fig 2. In situ hybridization at E16.5.** The femoral sections from *Runx2*$^{fl/fl}$ (A, D, G, J, M, P, S), *Runx2*$^{fl/+/Cre}$ (B, E, H, K, N, Q, T), and *Runx2*$^{fl/fl/Cre}$ (C, F, I, L, O, R, U) mice were used. (A-C) H–E staining. (D-U) In situ hybridization using *Col2a1* (D-F), *Col10a1* (G-I), *Mmp13* (J-L), *Ibsp* (M-O), *Spp1* (P-R), and *Col1a1* (S-U) probes. The boxed regions in A-U are magnified in A'-U', respectively. (V) Intensity on in situ hybridization. The gray values in the red boxes in G'-U' were measured. The gray values in *Runx2*$^{fl/fl}$ mice were set as 1, and the relative levels are shown. Scale bars: 100 μm. The number of mice analyzed: *Runx2*$^{fl/fl}$: 3–6, *Runx2*$^{fl/+/Cre}$: 3, *Runx2*$^{fl/fl/Cre}$: 3–5. Data are the mean ± SD. *Versus *Runx2*$^{fl/fl}$, #Versus *Runx2*$^{fl/+/Cre}$, *,#$p < 0.05$, **,##$p < 0.01$.

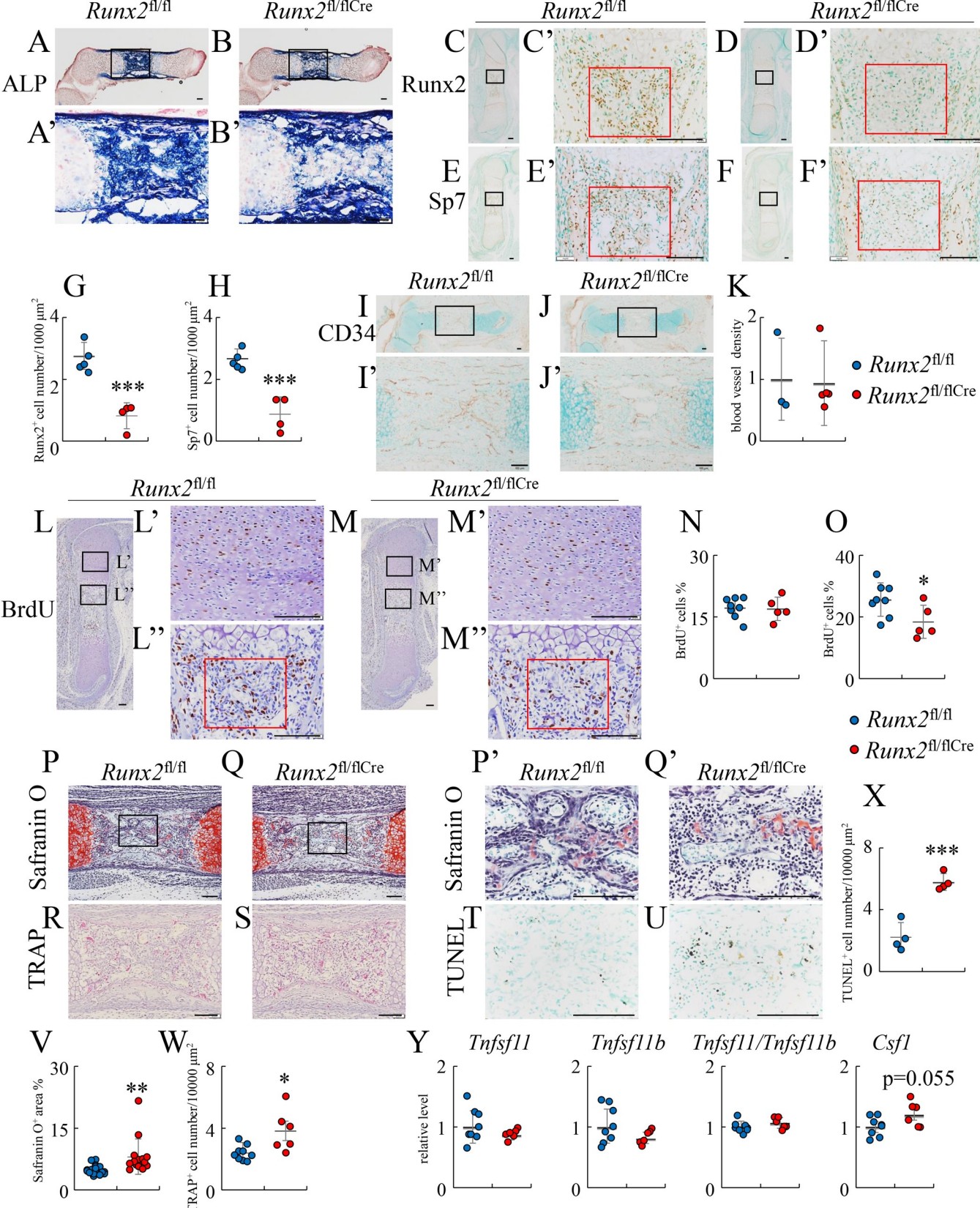

**Fig 3. Histological analyses of *Runx2*<sup>fl/fl</sup> and *Runx2*<sup>fl/fl/Cre</sup> embryos at E16.5.** The femoral sections from *Runx2*<sup>fl/fl</sup> (A, C, E, I, L, P, R, T) and *Runx2*<sup>fl/fl/Cre</sup> (B, D, F, J, M, Q, S, U) mice were used. (A and B) ALP staining. (C-F) Immunohistochemical analysis using anti-Runx2 (C, D) and anti-Sp7 (E, F) antibodies. The boxed regions in A-F were magnified in A'-F', respectively. (G and H) The numbers of Runx2 (G)- and Sp7 (H)-positive cells in the red boxes of C'-F' were

counted. (I-K) Immunohistochemical analysis using anti-CD34 antibody. The boxed regions in I and J were magnified in I' and J', respectively. The densities of stained endothelial cells were measured in the boxed region in I and J, the value in $Runx2^{fl/fl}$ mice was set as 1, and the relative levels are shown (K). The number of mice analyzed: $Runx2^{fl/fl}$: 3, $Runx2^{fl/fl\ Cre}$: 3. (L and M) BrdU labeling. The boxed regions in L were magnified in L' and L", and the boxed regions in M were magnified in M' and M". (N and O) Frequencies of BrdU-positive cells. The frequencies of BrdU-positive chondrocytes in the proliferating layers (L', M', N) and those of BrdU-positive cells in the primary spongiosa regions (red boxes in L" and M", O) were counted. (P and Q) Safranin O staining. The boxed regions in P and Q were magnified in P' and Q', respectively. (R and S) TRAP staining. (T and U) TUNEL staining. Serial sections were used in P and T, and in Q and U. (V) The percentage of the safranin O-positive area in bone marrow. (W) TRAP-positive cell number in bone marrow. (X) The number of TUNEL-positive cells. TUNEL-positive cells were counted in the regions that correspond to the boxed regions in P and Q using the serial sections. (Y) Real-time RT-PCR analysis of the genes related to osteoclastogenesis. RNA was extracted from the diaphysis of femurs and tibiae in $Runx2^{fl/fl}$ and $Runx2^{fl/fl/Cre}$ mice using razor blade and stereoscopic microscope at E16.5. The values in $Runx2^{fl/fl}$ mice were defined as 1, and the relative levels are shown. Scale bars: 100 μm. The number of mice analyzed: $Runx2^{fl/fl}$: 4–5, $Runx2^{fl/fl/Cre}$: 4 in ALP staining, immunohistochemical analysis, and TUNEL staining. $Runx2^{fl/fl}$: 8, $Runx2^{fl/fl/Cre}$: 5 in BrdU labeling. $Runx2^{fl/fl}$: 17, $Runx2^{fl/fl/Cre}$: 15 in safranin O staining. $Runx2^{fl/fl}$: 9, $Runx2^{fl/fl/Cre}$: 6 in TRAP staining. $Runx2^{fl/fl}$: 8, $Runx2^{fl/fl\ Cre}$: 6 in real-time RT-PCR analysis. Data are the mean ± SD. *Versus $Runx2^{fl/fl}$, *p<0.05, **p<0.01, ***p<0.001.

RNA from the diaphyses of femurs and tibiae at E16.5, which contained hypertrophic and terminal hypertrophic chondrocytes, osteoblasts, and hematopoietic cells. The expression of *Tnfsf11*, *Tnfrsf11b*, and *Csf1* and the ratio of *Tnfsf11/Tnfrsf11b* were similar between $Runx2^{fl/fl}$ and $Runx2^{fl/flCre}$ mice (Fig 3Y). However, *Tnfsf11* and *Csf1* expression and the ratio of *Tnfsf11/Tnfrsf11b* were reduced in the hypertrophic and terminal hypertrophic chondrocytes in $Runx2^{fl/flCre}$ mice at E15.5 (Fig 1R), which is consistent with the previous finding that Runx2 induces *Tnfsf11* expression [22]. Thus, it was speculated that *Tnfsf11* expression was upregulated in the invaded osteoblast lineage cells by the danger-associated molecular patterns (DAMPs) released from the necrotic terminal hypertrophic chondrocytes derived from the increased apoptotic terminal hypertrophic chondrocytes in $Runx2^{fl/flCre}$ mice. High mobility group box 1 (Hmgb1) is one of the DAMPs, which are released from necrotic cells and induce *Tnfsf11* expression in osteoblast lineage cells by promoting the production of proinflammatory cytokines, including tumor necrosis factor α (TNF-α), interleukin (IL)-6, and IL-1, in macrophages, dendritic cells, monocytes, and neutrophils [23]. Hmgb1 is normally localized in the nucleus and translocated to the cytosol with many types of cellular stress [24]. Thus, we examined the localization of Hmgb1 protein using femoral sections. Unexpectedly, Hmgb1 protein was dominantly localized in cytosol in hypertrophic and terminal hypertrophic chondrocytes in both $Runx2^{fl/fl}$ and $Runx2^{fl/flCre}$ mice, and the stained area and intensity were also similar between them (S2 Fig). The cytosolic Hmgb1 in the terminal hypertrophic chondrocytes will be immediately released when their necrosis occurs and the released Hmbg1 will induce *Tnfsf11* expression in osteoblast lineage cells.

## Vegfa expression was high in osteoblasts in the bone collar but severely reduced in the region of presumptive primary spongiosa in $Runx2^{fl/flCre}$ mice at E16.5

Runx2 has been reported to regulate Vegfa expression and be required for vascular invasion into the cartilage [8, 9]. However, vascular invasion into the cartilage was normal in $Runx2^{fl/flCre}$ mice (Figs 1F, 1G and 3I–3K). Thus, we examined Vegfa expression by immunohistochemistry using anti-Vegfa antibody at E16.5 (Fig 4). Osteoblasts in the bone collar of femurs strongly expressed Vegfa in both $Runx2^{fl/fl}$ and $Runx2^{fl/flCre}$ mice, whereas the expression of Vegfa was markedly reduced in the region of presumptive primary spongiosa due to the absence of osteoblasts in $Runx2^{fl/flCre}$ mice at E16.5 (Fig 4H, 4J, 4M, 4O, 4R and 4T).

## Transdifferentiation of terminal hypertrophic chondrocytes into osteoblasts was impaired in $Runx2^{fl/flCre}$ mice

To investigate whether the absence of osteoblasts in the region of presumptive primary spongiosa is due to the failure of transdifferentiation of terminal hypertrophic chondrocytes into

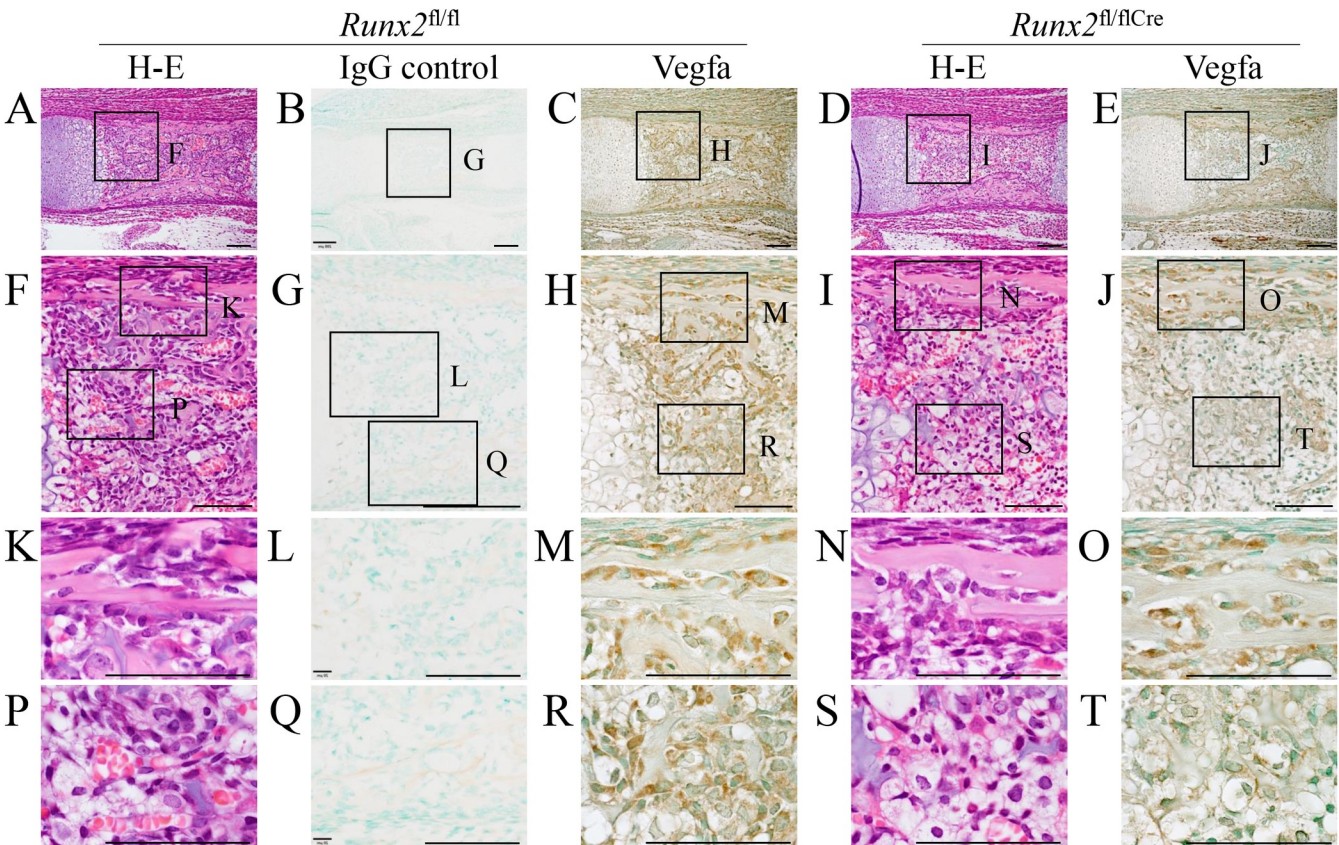

**Fig 4. Vegfa expression in *Runx2*<sup>fl/fl</sup> and *Runx2*<sup>fl/fl/Cre</sup> embryos at E16.5.** H–E staining (A, D, F, I, K, N, P, S) and immunohistochemical analysis using anti-Vegfa antibody (C, E, H, J, M, O, R, T) or same working concentration of normal mouse IgG (B, G, L, Q) of femoral sections from *Runx2*<sup>fl/fl</sup> (A-C, F-H, K-M, P-R) and *Runx2*<sup>fl/fl/Cre</sup> (D, E, I, J, N, O, S, T) mice. The boxed regions in A-E were magnified in F-J, respectively. The boxed regions in F-J were magnified in K and P, L and Q, M and R, and N and S, and O and T, respectively. Scale bars: 100 μm. The number of mice analyzed: *Runx2*<sup>fl/fl</sup>: 2, *Runx2*<sup>fl/fl/Cre</sup>: 2.

osteoblasts, *Runx2*<sup>fl/flCre</sup> mice were mated with CAG promoter-LacZ reporter mice to generate *Runx2*<sup>fl/+Cre LacZ</sup> and *Runx2*<sup>fl/flCre LacZ</sup> mice. In both *Runx2*<sup>fl/+Cre LacZ</sup> and *Runx2*<sup>fl/flCre LacZ</sup> mice at E15.5, hypertrophic chondrocytes and terminal hypertrophic chondrocytes on femoral sections were stained by β-galactosidase (Fig 5A–5H, S3A Fig). Before vascular invasion into the cartilage, Col1a1-positive cells were present in the bone collar, but not in the terminal hypertrophic chondrocyte layer of femurs, in both *Runx2*<sup>fl/+Cre LacZ</sup> and *Runx2*<sup>fl/flCre LacZ</sup> mice, suggesting that the transdifferentiation of terminal hypertrophic chondrocytes into osteoblasts does not occur before vascular invasion into the cartilage (S3D and S3E Fig). At E16.5 and E17,5, the number of β-galactosidase-positive cells in the region of primary spongiosa in *Runx2*<sup>fl/flCre LacZ</sup> mice was significantly lower than that in *Runx2*<sup>fl/+Cre LacZ</sup> mice (Fig 5I–5U, S3B and S3C Fig). The β-galactosidase-positive cells in *Runx2*<sup>fl/+Cre LacZ</sup> mice were morphologically terminal hypertrophic chondrocytes or osteoblast lineage cells, whereas those in *Runx2*<sup>fl/flCre LacZ</sup> mice were morphologically terminal hypertrophic chondrocytes (Fig 5Q–5T). These findings indicate that the absence of primary spongiosa and osteoblasts in the region of presumptive primary spongiosa in *Runx2*<sup>fl/flCre LacZ</sup> mice was due to the impaired transdifferentiation of terminal hypertrophic chondrocytes into osteoblasts.

To more precisely evaluate the transdifferentiation of chondrocytes into osteoblasts, we mated *Runx2*<sup>fl/flCre</sup> mice with Rosa26-CAG-loxP-mTFP1 (Rosa) reporter mice [25] and 2.3 kb *Cola1* promoter-tdTomato (tomato) mice to generate *Runx2*<sup>fl/+Cre Rosa tomato</sup> mice

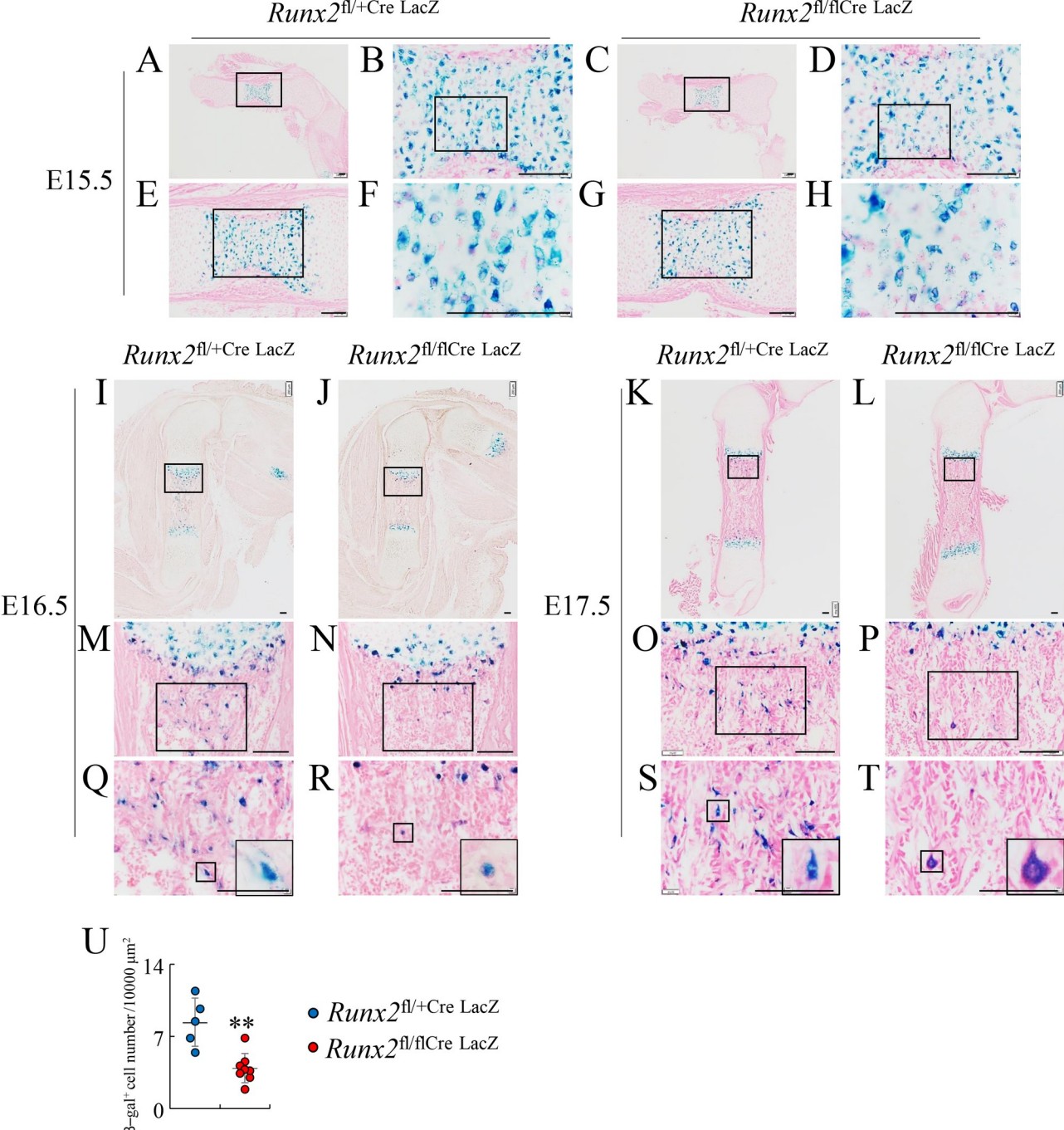

**Fig 5. β-galactosidase staining of femoral sections from *Runx2*<sup>fl/+/Cre LacZ</sup> and *Runx2*<sup>fl/fl/Cre LacZ</sup> mice at E15.5, E16.5, and E17.5.** β-galactosidase staining of femoral sections from *Runx2*<sup>fl/+/Cre LacZ</sup> (A, I, K) and *Runx2*<sup>fl/fl/Cre LacZ</sup> (C, J, L) mice at E15.5 (A, C), E16.5 (I, J), and E17.5 (K, L). The boxed regions in A, E, B, C, G, D, I, M, J, N, K, O, L, and P were magnified in E, B, F, G, D, H, M, Q, N, R, O, S, P, and T, respectively. The cells in the boxes in Q, R, S, and T were magnified in the windows. (U) The number of β-gal positive cells in Q and R. Scale bars: 100 μm. The number of mice analyzed: *Runx2*<sup>fl/+/Cre LacZ</sup>: 2, *Runx2*<sup>fl/fl/Cre LacZ</sup>: 3 at E15.5. *Runx2*<sup>fl/+/Cre LacZ</sup>: 5, *Runx2*<sup>fl/fl/Cre LacZ</sup>: 8 at E16.5. *Runx2*<sup>fl/+/Cre LacZ</sup>: 4, *Runx2*<sup>fl/fl/Cre LacZ</sup>: 2 at E17.5. Data are the mean ± SD. *Versus *Runx2*<sup>fl/+/Cre LacZ</sup>, **p<0.01.

and *Runx2*<sup>fl/flCre Rosa tomato</sup> mice. The 2.3 kb *Cola1* promoter-tdTomato mice, which we newly generated, showed the tdTomato expression specifically in osteoblasts (S4 Fig). In

$Runx2^{\text{fl/+Cre Rosa tomato}}$ mice and $Runx2^{\text{fl/flCre Rosa tomato}}$ mice, osteoblasts showed red fluorescence (tomato$^+$), the cells derived from hypertrophic chondrocytes showed teal (blue-green) fluorescence (mTFP$^+$), and the osteoblasts derived from hypertrophic chondrocytes showed whitish fluorescence (tomato$^+$mTFP$^+$) on femoral sections by confocal microscope (Figs 6 and 7). We counted the number of these cells and calculated the percentages of the osteoblasts derived from hypertrophic chondrocytes in total osteoblasts (tomato$^+$mTFP$^+$/tomato$^+$). About 30%, 20%, and 15% of osteoblasts in the trabecular bone was derived from chondrocytes in control $Runx2^{\text{fl/+Cre Rosa tomato}}$ mice at E17.5, newborn, and 1 week of age, respectively, whereas they were 1–3% in $Runx2^{\text{fl/flCre Rosa tomato}}$ mice (Fig 6). In the endosteum of cortical bone at 1 week of age, about 4% of osteoblasts was derived from chondrocytes in $Runx2^{\text{fl/+Cre Rosa tomato}}$ mice but they were absent in $Runx2^{\text{fl/flCre Rosa tomato}}$ mice (Fig 7A–7G). At 3 weeks of age, about 15% of osteoblasts in the trabecular bone and in the endosteum of cortical bone in $Runx2^{\text{fl/+Cre Rosa tomato}}$ mice was derived from chondrocytes (Fig 7H–7Q). Further, osteoblasts derived from chondrocytes were present in the secondary ossification center of femurs in $Runx2^{\text{fl/+Cre LacZ}}$ mice but they were virtually absent in that in $Runx2^{\text{fl/flCre LacZ}}$ mice at 3 weeks of age (S5A–S5C Fig).

## The primary spongiosa was formed along with the appearance of osteoblasts in the region in $Runx2^{\text{fl/flCre}}$ mice, but it was less developed than that in $Runx2^{\text{fl/fl}}$ mice at E17.5 and the newborn stage

To investigate the physiological effects of the impaired transdifferentiation, bone development was examined in $Runx2^{\text{fl/flCre}}$ mice at E17.5 and the newborn stage. Although the cartilage matrix stained with safranin O was similarly observed in the region of the primary spongiosa in femurs of $Runx2^{\text{fl/flCre}}$ mice and $Runx2^{\text{fl/fl}}$ mice at E17.5, *Ibsp*, *Col1a1*, and *Bglap2* expression was significantly lower and *Spp1* expression was marginally lower in the region of the primary spongiosa in femurs or tibiae in $Runx2^{\text{fl/flCre}}$ mice than in $Runx2^{\text{fl/fl}}$ mice (Fig 8A–8Q), demonstrating that the primary spongiosa is less developed in $Runx2^{\text{fl/flCre}}$ mice than in $Runx2^{\text{fl/fl}}$ mice. Immunohistochemical analysis using femoral sections revealed Runx2-positive osteoblasts and Sp7-positive osteoblasts in the region of primary spongiosa in $Runx2^{\text{fl/flCre}}$ mice, but their number was less than that in $Runx2^{\text{fl/fl}}$ mice at E17.5 (Fig 8R–8W).

At the newborn stage, primary spongiosa developed further, and the number of Runx2-positive cells and the expression of *Col1a1* and *Bglap2* in the primary spongiosa in femurs of $Runx2^{\text{fl/flCre}}$ mice reached a similar level to those of $Runx2^{\text{fl/fl}}$ mice (Fig 9A–9K). The physical appearance and body weight were similar between $Runx2^{\text{fl/fl}}$ and $Runx2^{\text{fl/flCre}}$ newborn mice (Fig 9L, 9M and 9R). On micro-CT analysis, the total bone volume and total trabecular bone volume of femurs were similar between $Runx2^{\text{fl/fl}}$ and $Runx2^{\text{fl/flCre}}$ newborn mice (Fig 9N–9Q, 9S and 9T). However, the bone volume of the secondary spongiosa in $Runx2^{\text{fl/flCre}}$ newborn mice was less than that in $Runx2^{\text{fl/fl}}$ newborn mice (Fig 9P, 9Q and 9U).

## The trabecular bone volume reached a normal level in $Runx2^{\text{fl/flCre}}$ mice by 6 weeks of age

The physical appearance and body weight were similar between $Runx2^{\text{fl/fl}}$ and $Runx2^{\text{fl/flCre}}$ females at 6 weeks and males at 20 weeks of age (Fig 10A–10C and 11A–11C). Micro-CT analysis demonstrated that all of the parameters of trabecular bone in the primary and secondary ossification centers, including bone volume, trabecular thickness, trabecular number, and trabecular bone mineral density (BMD), were similar in femurs between $Runx2^{\text{fl/fl}}$ and $Runx2^{\text{fl/flCre}}$ mice at 6 weeks and 20 weeks of age (Fig 10D–10G, 10L, 10M, 11D–11G, 11L and 11M).

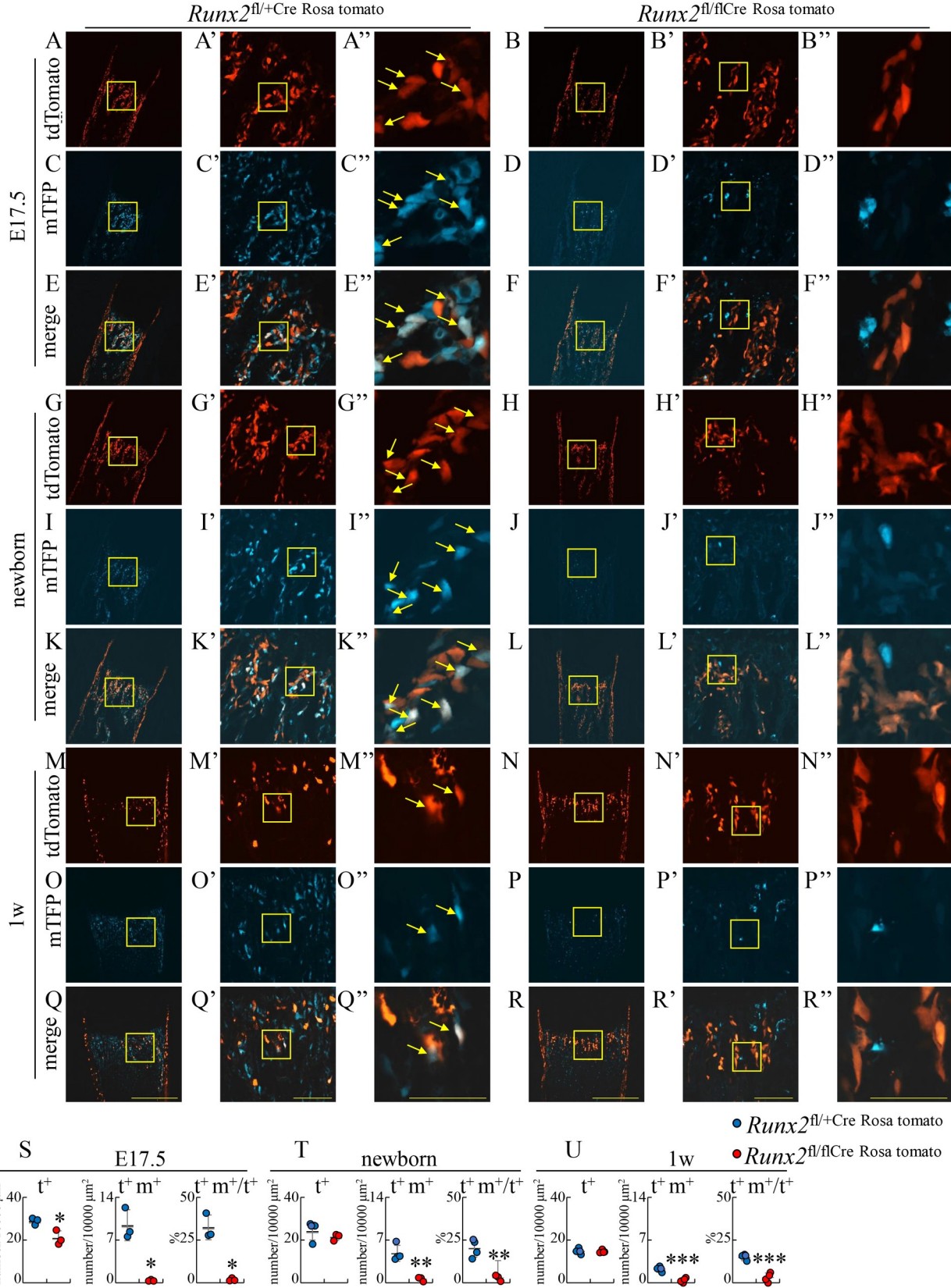

**Fig 6. Tracing of the transdifferentiated osteoblasts in trabecular bone at E17.5, newborn, and one week of age.** Frozen femoral sections from *Runx2*fl/+Cre ROSA tomato (A, C, E, G, I, K, M, O, Q) and *Runx2*fl/flCre ROSA tomato (B, D, F, H, J, L, N, P, R) mice at E17.5 (A-F), newborn (G-L), and 1 week of age (M-R) were observed by confocal microscopy. Osteoblasts express tdTomato (red), the cells derived from hypertrophic chondrocytes express mTFP (blue-green), and the osteoblasts derived from hypertrophic chondrocytes exhibit their merged whitish color. The boxed regions in A-R and A'-R' are magnified in A'-R' and A''-R'', respectively. Arrows show the osteoblasts derived from hypertrophic chondrocytes. Scale bars: 500 μm (A-R), 100μm (A'-R'), and 50μm (A''-R''). (S-U) The number of tdTomato-positive ($t^+$) cells and tdTomoto- and mTFP-double positive ($t^+m^+$) cells were counted in trabecular bone, and the percentages of $t^+m^+$ cells in $t^+$ cells were calculated at E17.5 (S), newborn (T), and 1 week of age (U). Arrows show $t^+m^+$ cells. The number of mice analyzed: *Runx2*fl/+Cre Rosa tomato: 3–4, *Runx2*fl/flCre Rosa tomato: 3–4. Data are the mean ± SD. *Versus *Runx2*fl/+ Cre Rosa tomato, *$p<0.05$, **$p<0.01$, ***$p<0.001$.

The parameters of cortical bone, including cortical volume (CtAr/TtAr), cortical thickness (Ct. Th), and cortical BMD (BMD), in femurs were also similar between *Runx2*fl/fl and *Runx2*fl/flCre mice at 6 weeks and 20 weeks of age (Fig 10H, 10I, 10N, 11H, 11I and 11N). Furthermore, the parameters of trabecular bones in the first lumbar vertebrae were similar between *Runx2*fl/fl and *Runx2*fl/flCre mice at 6 weeks and 20 weeks of age (Fig 10J, 10K, 10O, 11J, 11K and 11O).

In bone morphometric analysis of cortical bone at 6 weeks and 20 weeks of age, the mineral apposition rate, mineralizing surface, and bone formation rate were similar between *Runx2*fl/fl and *Runx2*fl/flCre mice in both the endosteum and periosteum (Fig 10P–10W and 11P–11W). We also preformed bone histomorphometric analysis of trabecular bone of vertebrae in females at 6 weeks and 12 weeks of age and tibiae in females at 12 weeks of age. All parameters for osteoblasts, osteoclasts, and bone resorption and formation were similar between *Runx2*fl/fl and *Runx2*fl/flCre mice (Fig 12). Finally, we performed three-point bending tests to evaluate the bone strength using the female femurs at 12 weeks of age. The parameters, including maximum load, displacement, stiffness, and energy to failure, were similar between *Runx2*fl/fl and *Runx2*fl/flCre mice (Fig 13).

## Discussion

The process of endochondral ossification, including terminal differentiation of chondrocytes and vascular invasion into the cartilage, was normal in *Runx2*fl/flCre mice, although the expression of *Mmp13*, *Ibsp*, *Spp1*, and *Vegfa* in the terminal hypertrophic chondrocytes was reduced. However, transdifferentiation of the terminal hypertrophic chondrocytes into osteoblasts was impaired in *Runx2*fl/flCre mice, and apoptosis of terminal hypertrophic chondrocytes was increased in *Runx2*fl/flCre mice before the transdifferentiation in *Runx2*fl/fl mice. The osteoblasts derived from chondrocytes were about 15–30% of total osteoblasts in the trabecular bone and 4–15% in the endosteum of cortical bone in control *Runx2*fl/+Cre mice, whereas those were about 1–3% in the trabecular bone and absent in the cortical bone in *Runx2*fl/flCre mice (Figs 6 and 7). *Runx2*fl/flCre mice lacked osteoblasts in the region of presumptive primary spongiosa and failed to form the primary spongiosa at E16.5. However, osteoblasts, which likely originated from the perichondrium/periosteum, were observed in the region of presumptive primary spongiosa in *Runx2*fl/flCre mice at E17.5, the number of osteoblasts became similar to that in *Runx2*fl/fl mice at birth, and the bone volume of the primary spongiosa but not secondary spongiosa became similar. In addition, all of the bone parameters, including trabecular and cortical bone, became similar between *Runx2*fl/flCre and *Runx2*fl/fl mice at 6 weeks of age, and the bone was maintained thereafter acquiring normal strength. Thus, our findings demonstrate that Runx2 is required for the survival of terminal hypertrophic chondrocytes and their transdifferentiation into osteoblasts, that this transdifferentiation is transiently required for forming the primary spongiosa, and that osteoblasts, which have most likely originated from the perichondrium/periosteum, can compensate for the lack of transdifferentiation, leading to normal bone mass in *Runx2*fl/flCre mice at 6 weeks of age.

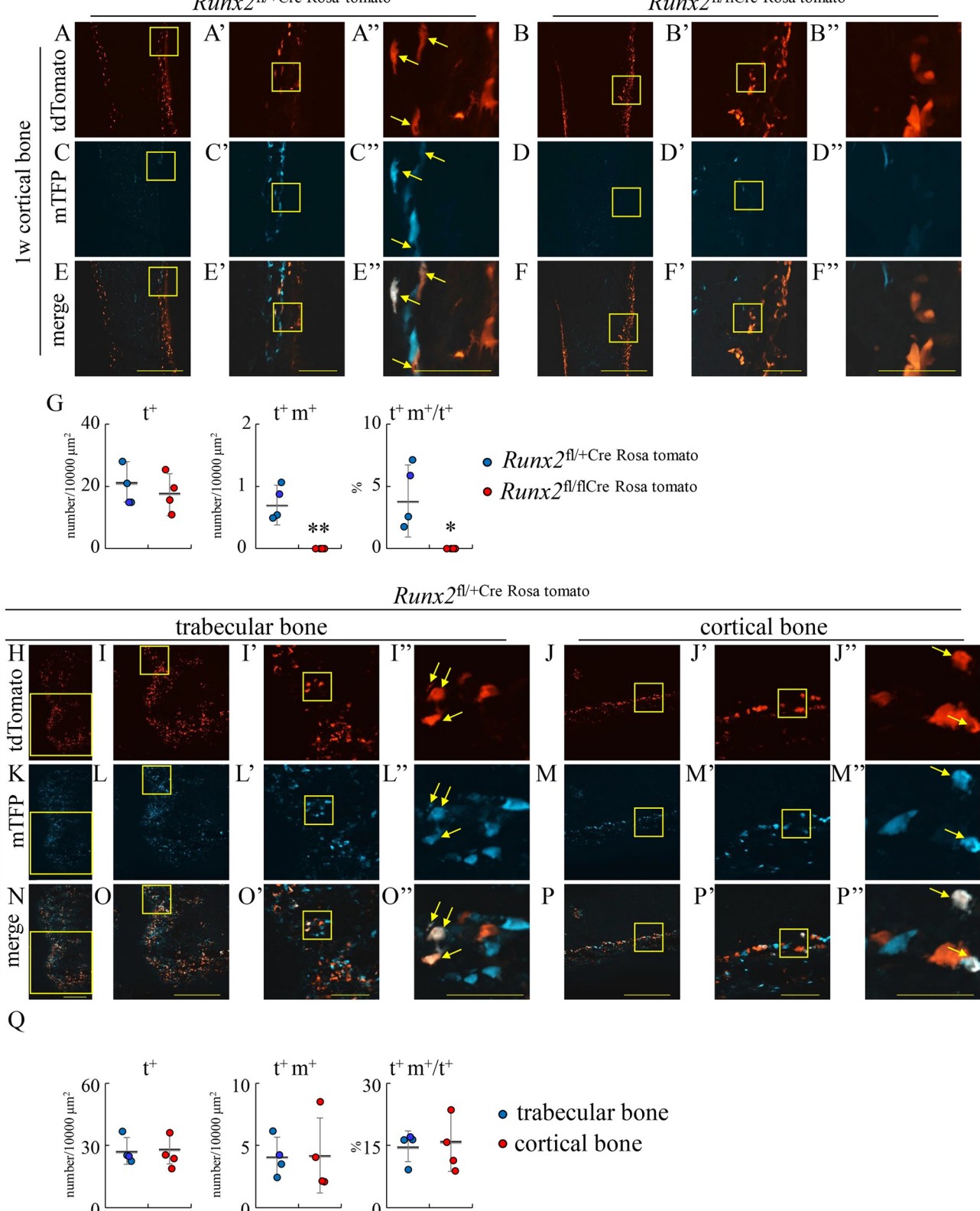

**Fig 7. Tracing of the transdifferentiated osteoblasts in cortical bone at 1 week of age, and trabecular and cortical bone at 3 weeks of age.** (A-G) Cortical bone in frozen femoral sections in *Runx2*<sup>fl/+Cre Rosa tomato</sup> (A, C, E) and *Runx2*<sup>fl/flCre Rosa tomato</sup> (B, D, F) mice at 1 week of age. The boxed regions in A-F and

A'-F' were magnified in A'-F' and A"-F", respectively. The number of tdTomato-positive (t$^+$) cells and tdTomoto- and mTFP-double positive (t$^+$m$^+$) cells were counted in the endosteum of cortical bone, and the percentages of t$^+$m$^+$ cells in t$^+$ cells were calculated (G). (H-P) Trabecular (H, K, N) and cortical (J, M, P) bone in the frozen femoral sections in *Runx2*$^{\text{fl/+Cre Rosa tomato}}$ mice at 3 weeks of age. The boxed region in H, I, I', J, J', K, L, L', M, M', N, O, O', P, and P' were magnified in I, I', I", J', J", L, L', L", M', M", O, O', O", P', and P", respectively. The number of tdTomato-positive (t$^+$) cells and tdTomoto- and mTFP-double positive (t$^+$m$^+$) cells were counted in the trabecular bone and in the endosteum of cortical bone, and the percentages of t$^+$m$^+$ cells in t$^+$ cells were calculated (Q). Arrows show t$^+$m$^+$ cells. Scale bars: 1mm (H, K, N), 500 μm (A-F, I, J, L, M, O, P), 100 μm (A'-F', I', J', L', M', O', P'), and 50 μm (A"-F", I", J", L", M", O", P"). The number of mice analyzed: *Runx2*$^{\text{fl/+Cre Rosa tomato}}$: 3–4, *Runx2*$^{\text{fl/flCre Rosa tomato}}$: 3–4. $^*$Versus *Runx2*$^{\text{fl/+Cre Rosa tomato}}$, $^*$p<0.05, $^{**}$p<0.01.

*Runx2*$^{-/-}$ mice die at birth and have cartilaginous skeletons due to the impaired chondrocyte maturation and the absence of osteoblasts [2, 5–7]. In most skeletons of *Runx2*$^{-/-}$ mice, hypertrophic chondrocytes are absent. However, maturation of hypertrophic chondrocytes into terminal hypertrophic chondrocytes normally proceeded in *Runx2*$^{\text{fl/flCre}}$ mice, although the terminal hypertrophic chondrocytes expressed *Mmp13*, *Ibsp*, *Spp1*, and *Vegfa*, which are target genes of Runx2 [26], at reduced levels as compared with those in *Runx2*$^{\text{fl/fl}}$ mice. Thus, Runx2 is indispensable for the expression of *Mmp13*, *Ibsp*, *Spp1* and *Vegfa* at the physiological levels in terminal hypertrophic chondrocytes but dispensable for the final maturation of hypertrophic chondrocytes into terminal hypertrophic chondrocytes.

In the restricted skeletons of *Runx2*$^{-/-}$ mice, including the tibia, fibula, radius, and ulna, the cartilage matrix is mineralized and chondrocytes are terminally differentiated [2, 5]. However, vascular invasion into the calcified cartilage is absent in *Runx2*$^{-/-}$ mice and Vegfa expression in the terminal hypertrophic chondrocytes is reduced [8, 9]. Therefore, Runx2 has been considered to induce the vascular invasion into the cartilage by regulating Vegfa expression in terminal hypertrophic chondrocytes [27]. Vegfa conditional knockout mice using *Col2a1*-Cre, which directs the transgene expression to chondrocytes and perichondral cells, showed delayed vascular invasion into the cartilage [28]. The delayed vascular invasion is due to the lack of Vegfa expression in the terminal hypertrophic chondrocytes and perichondrial osteoblasts, both of which strongly express Vegfa. Similarly, Vegfa conditional knockout mice using *Sp7*-Cre, which directs the transgene expression to prehypertrophic chondrocytes and preosteoblasts in the perichondrium, showed delayed vascular invasion into the cartilage [29]. The delayed vascular invasion is also due to the lack of Vegfa expression in terminal hypertrophic chondrocytes and perichondrial osteoblasts. Vegfa expression in the terminal hypertrophic chondrocytes was reduced in *Runx2*$^{-/-}$ mice and *Runx2*$^{\text{fl/flCre}}$ mice, but vascular invasion into the cartilage never occurs in *Runx2*$^{-/-}$ mice but normally occurred in *Runx2*$^{\text{fl/flCre}}$ mice (Fig 1F, 1G and 1R, Fig 3I–3K) [2, 5–8]. The difference of the two mouse models is the presence of perichondrial osteoblasts in *Runx2*$^{\text{fl/flCre}}$ mice but their absence in *Runx2*$^{-/-}$ mice (Fig 4H, 4J, 4M and 4O) [6, 7]. These findings indicate that Vegfa expression in perichondrial osteoblasts is sufficient for the vascular invasion into the cartilage. The importance of the perichondrium for the vascular invasion into the cartilage was also shown by ex vivo experiments transplanting cartilaginous anlagen of long bones into the renal capsule of adult mice [30]. However, the increased apoptosis of terminal hypertrophic chondrocytes in *Runx2*$^{\text{fl/flCre}}$ mice may have promoted the vascular invasion into the cartilage because apoptotic terminal hypertrophic chondrocytes may finally die by necrosis without phagocytosis and release DAMPs, which induce osteoclastogenesis [23, 31]. Indeed, there were more osteoclasts in *Runx2*$^{\text{fl/flCre}}$ mice than in *Runx2*$^{\text{fl/fl}}$ mice at E16.5, although *Tnfsf11* and *Csf1* expression was reduced in the terminal hypertrophic chondrocytes (Figs 1R, 3R, 3S and 3W). The increased osteoclasts may also have facilitated the invasion of preosteoblasts in the perichondrium into the cartilage by removing the cartilage matrix. Furthermore, vascular invasion into the cartilage is likely required for the

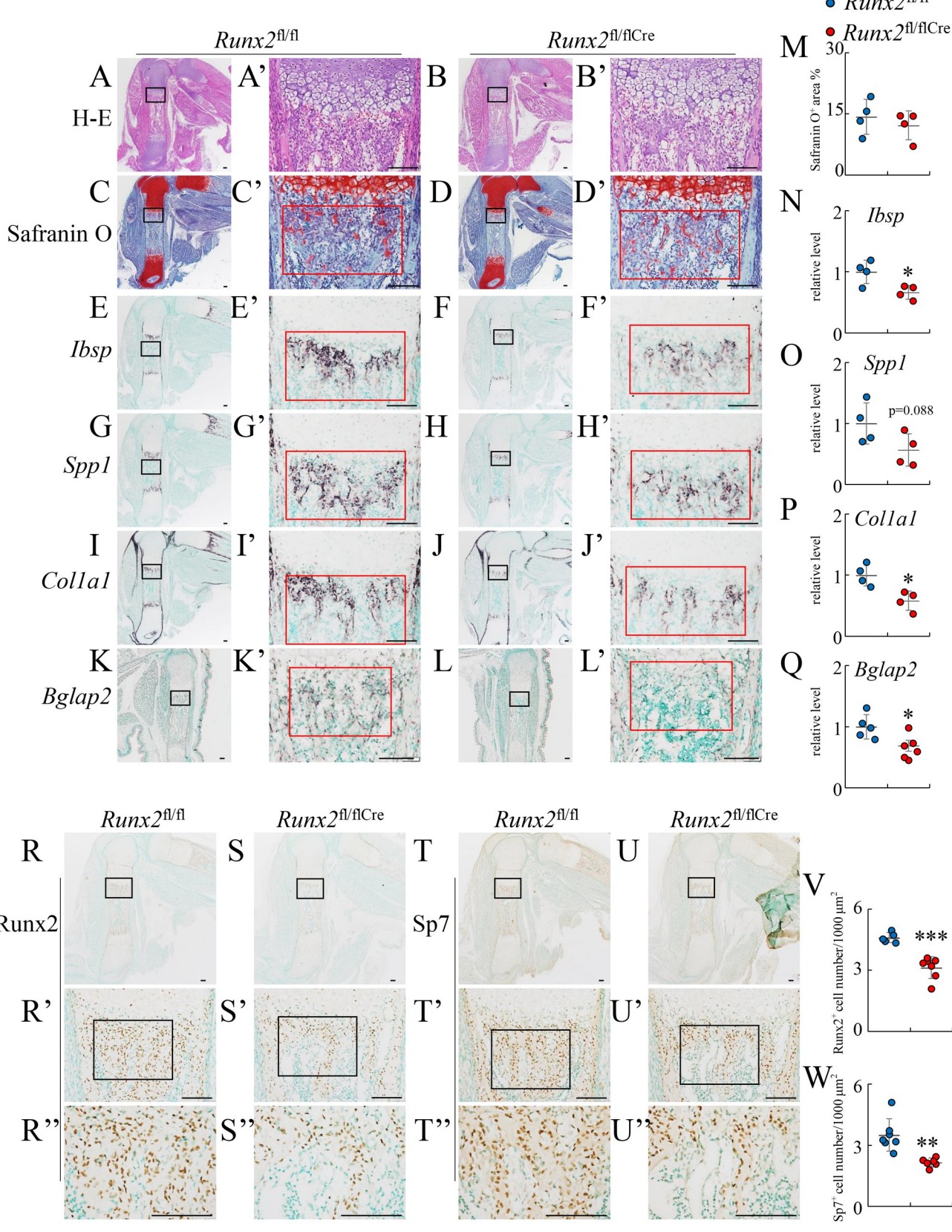

**Fig 8. Histological analyses of *Runx2*<sup>fl/fl</sup> and *Runx2*<sup>fl/fl/Cre</sup> embryos at E17.5.** Femoral (A-J, R-U) and tibial (K, L) sections from *Runx2*<sup>fl/fl</sup> (A, C, E, G, I, K, R, T) and *Runx2*<sup>fl/fl/Cre</sup> (B, D, F, H, J, L, S, U) mice were used for the analyses. (A and B) H–E staining. (C and D) Safranin O staining. (E–L) In situ hybridization using *Ibsp* (E, F), *Spp1* (G, H), *Col1a1* (I, J), and *Bglap2* (K, L) probes. (M) The percentage of the safranin O-positive area in red boxes of C' and D'. (N–Q) Intensity of in situ hybridization using *Ibsp*, *Spp1*, *Col1a1*, and *Bglap2* probes. The gray values in the red boxes in E'–L' were measured. The gray values in *Runx2*<sup>fl/fl</sup> mice were set as 1, and the relative levels are shown. (R-U) Immunohistochemical analysis using anti-Runx2 (R, S) and anti-Sp7 (T, U) antibodies. The boxed regions in A-L, R-U, and R'-U' were magnified in A'-L', R'-U', and R"-U", respectively. Scale bars: 100 µm. (V, W) The number of Runx2-positive cells in R" and S" (V) and Sp7-positive cells in T" and U" (W). The number of mice analyzed: *Runx2*<sup>fl/fl</sup>: 4–7, *Runx2*<sup>fl/fl/Cre</sup>: 4–7. Data are the mean ± SD. *Versus *Runx2*<sup>fl/fl</sup>, *p<0.05.

transdifferentiation of terminal hypertrophic chondrocytes because it was not observed before vascular invasion.

The major source of bone marrow osteoblasts has long been debated. The transdifferentiation of chondrocytes to osteoblasts was confirmed using *Col10a1* Cre, which is expressed in hypertrophic chondrocytes, but not in perichondrial/periosteal cells [16–19]. Furthermore, Pthlh-positive chondrocytes in the resting zone of the growth plate were reported to become osteoblasts and stromal cells in bone marrow, and to be able to differentiate to chondrocytes, osteoblasts, and adipocytes *in vitro* [32]. These findings suggest that the transdifferentiated stroma cells and osteoblasts are major sources of osteoblasts in bone marrow, and play an important role in trabecular and cortical bone formation. Cell-lineage tracing experiments using *Sp7* CreER, *Col2a1* CreER, *Sox9* CreER, *Acan* CreER, which are expressed in perichondrial/periosteal cells and chondrocytes, indicate that the origin of osteoblasts in bone marrow is perichondrial/periosteal cells and/or chondrocytes [33, 34]. Therefore, the osteoblasts in the primary spongiosa in *Runx2*<sup>fl/flCre</sup> mice after E17.5 must have originated from perichondrial/periosteal cells. Although the primary spongiosa formation in *Runx2*<sup>fl/flCre</sup> mice was similar to that in *Runx2*<sup>fl/fl</sup> mice at birth, the secondary spongiosa formation in *Runx2*<sup>fl/flCre</sup> mice was markedly retarded. The structure and volume of trabecular bone in *Runx2*<sup>fl/flCre</sup> mice became similar to those in *Runx2*<sup>fl/fl</sup> mice by 6 weeks of age. Thus, our findings indicate that chondrocyte transdifferentiation is required for trabecular bone formation at embryonic and neonatal stages, but dispensable for acquiring normal bone mass and structure in young and adult mice, and that osteoblasts originating from perichondrial/periosteal cells play a major role in trabecular and cortical bone formation at least after birth. It was reported that borderline chondrocytes, which are located adjacent to the perichondrium, differentiate to osteoblasts and stromal cells [35]. Although the borderline chondrocytes express Col10a1 before differentiation to osteoblasts [35], *Runx2*<sup>fl/flCre Rosa tomato</sup> mice had very low number of osteoblasts derived from chondrocytes. Therefore, the differentiation of borderline chondrocytes to osteoblasts was considered to have been impaired in *Runx2*<sup>fl/flCre</sup> mice.

In conclusion, Runx2 maintained the survival of terminal hypertrophic chondrocytes and was required for their transdifferentiation into osteoblasts. This transdifferentiation was required for trabecular bone formation in embryonic and neonatal stages, but osteoblasts originating from the perichondrium/periosteum were considered to play a major role in trabecular bone formation at least after birth. The molecular mechanism of chondrocyte transdifferentiation to osteoblasts needs to be further investigated.

## Materials and methods

### Ethics statement

Before the study, all experimental protocols were reviewed and approved by the Animal Care and Use Committee of Nagasaki University Graduate School of Biomedical Sciences (No. 1903131520–2).

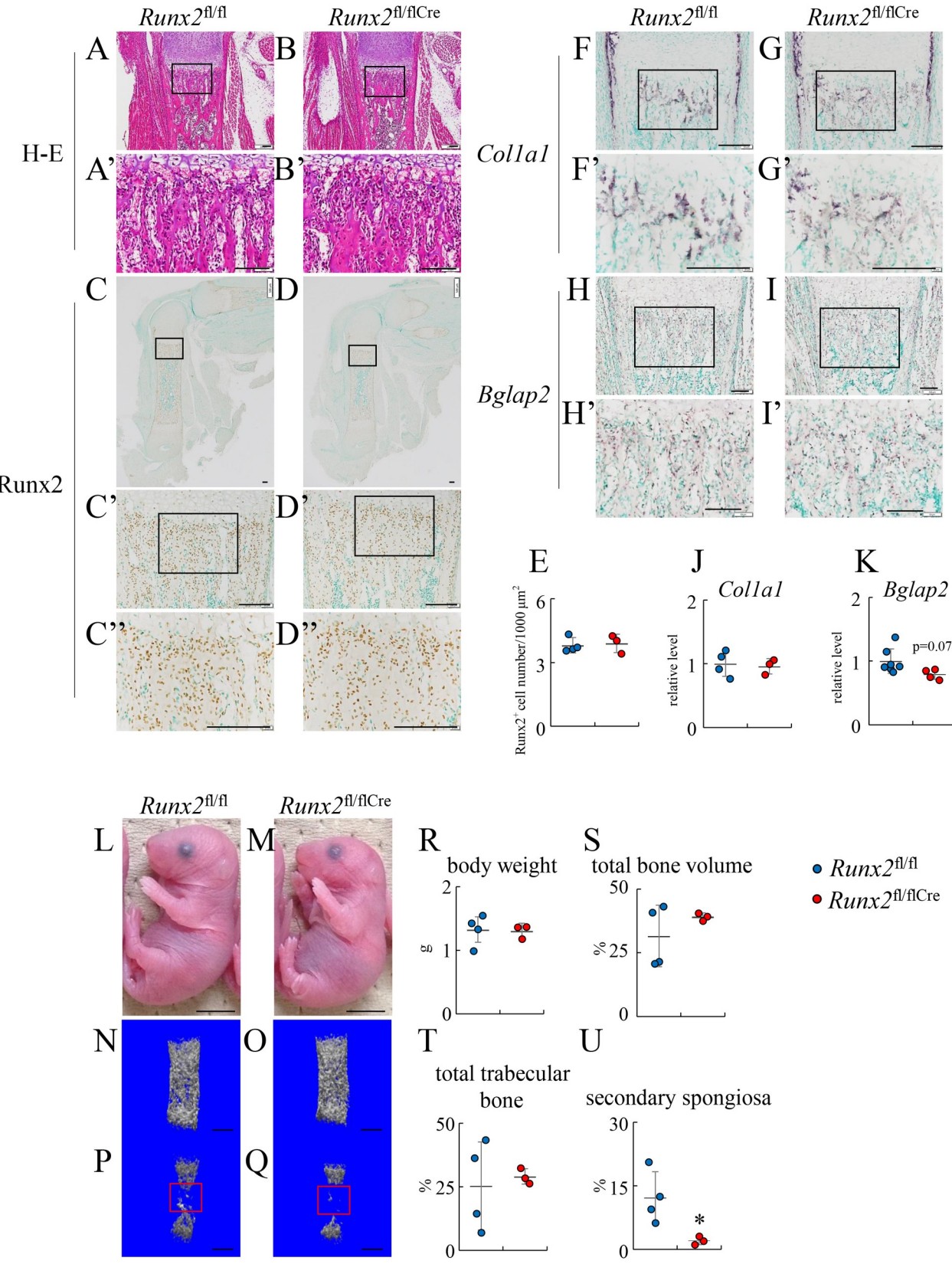

**Fig 9. Histological and micro-CT analyses of *Runx2*<sup>fl/fl</sup> and *Runx2*<sup>fl/flCre</sup> newborn mice.** The femoral sections from *Runx2*<sup>fl/fl</sup> (A, C, F, H) and *Runx2*<sup>fl/fl/Cre</sup> (B, D, G, I) mice were used for histological analyses. (A and B) H–E staining. (C and D) Immunohistochemical analysis using anti-Runx2 antibody. (E) The number of Runx2-positive cells in C" and D". (F-I) In situ hybridization using *Col1a1* and *Bglap2* probes. (J and K) Intensity of in situ hybridization using the *Col1a1* probe in F' and G'(J) and *Bglap2* probe in H' and I' (K). The values in *Runx2*<sup>fl/fl</sup> mice were set as 1, and the relative levels are shown. (L, M) Physical appearance of *Runx2*<sup>fl/fl</sup> and *Runx2*<sup>fl/flCre</sup> mice. (N-Q) Micro-CT images of whole femurs (N, O) and the trabecular bone (P, Q) in *Runx2*<sup>fl/fl</sup> (N, P) and *Runx2*<sup>fl/flCre</sup> (O, Q) mice. (R) Body weight of *Runx2*<sup>fl/fl</sup> and *Runx2*<sup>fl/flCre</sup> mice. (S-U) The percentage of the total bone volume/total tissue volume in N and O (S), that of total trabecular bone volume/total tissue volume in P and Q (T), and that of secondary spongiosa volume/tissue volume in the red boxes in P and Q (U). The boxed regions in A, B, C, D, C', D', F, G, H, and I are magnified in A', B', C', D', C", D", F', G', H', and I', respectively. Scale bars: 100 μm (A-D", F-I'), 0.5 cm (L, M), 0.5 mm (N-Q). The number of mice analyzed: *Runx2*<sup>fl/fl</sup>: 4–7, *Runx2*<sup>fl/fl/Cre</sup>: 3–4. Data are the mean ± SD. *Versus *Runx2*<sup>fl/fl</sup>, *p<0.05.

## Generation of *Runx2*<sup>fl/fl/Cre</sup> mice and 2.3 kb *Cola1* promoter-tdTomato transgenic mice

Exon 3 of *Runx2* previously described as exon 1 [6], which encodes the N-terminal region and a part of Runt domain of Runx2 protein was conditionally deleted. The targeting vector harboring the exon 3 and a FRT-flanked neomycin resistance gene (Neo), both of which are flanked by Loxp sites was electroporated into Bruce-4 ES cells (C57BL/6). Digested genomic DNA of targeted ES cells was subjected to Southern blot using 5' or 3' alkaline phosphatase-labelled probes (429bp or 387bp), which were generated by polymerase chain reaction (PCR) using 5'-GATAGAATGATTTCAAGGAGGAGTG-3' and 5'-CTTGTAACTGCGTTATGAAA CTTGA-3' or 5'-AAGGAGACTTTTAAGCTTCAGACAT-3' and 5'-AAAGGATTCCTGAG AGTGAAATACA-3' primers, respectively. To remove the FRT-flanked Neo, the offspring (F1) was crossed with CAG-FLP transgenic mice (C57BL/6). Floxed and wild-type alleles of *Runx2*<sup>flox/+</sup> mice were detected by PCR using 5'-ACTTCGGTTGGTCAGAATAAACAG-3' and 5'-CAAGCTAACGGGACTTG GAAGAG-3' primers (327 bp and 173 bp, respectively). *Runx2*<sup>fl/fl</sup> mice were mated with *Col10a1*-Cre transgenic mice [21] to generate *Runx2*<sup>fl/fl/Cre</sup> mice. The background of *Runx2*<sup>fl/fl</sup> mice and *Col10a1*-Cre transgenic mice was C57BL/6. To generate 2.3k *Col1a1* promoter-tdTomato transgenic mice, a 1456-bp tdTomato (Addgene, Watertown, CM) fragment was cloned into the NotI site of a pNASSβ expression vector (Clontech, Mountain View, CA), which contained the 2.3-kb *Col1a1* promoter region. The construct DNA was injected into the pronuclei of fertilized eggs from C57BL/6 mice. Transgenic mice were identified by genomic PCR using 5'-ATGCGGCCGCATCCACCGGTCGCCACCAT-3' and 5'-ATGCGGCCGCTTACTTGTACAGCT-3' primers. A transgenic line was maintained in the C57BL/6 background. Rosa26-CAG-loxP-mTFP1 reporter mice with C57BL/6 background were obtained from RIKEN Bioresource Research Center [25]. Animals were housed 3 per cage in a pathogen-free environment on a 12-h light cycle at 22±2˚C with standard chow (CLEA Japan, Tokyo, Japan) and had free access to tap water.

## Histological analyses

For histological analyses, mice were fixed in 4% paraformaldehyde/0.1 M phosphate buffer, and embedded in paraffin. Sections of 4 or 7 μm in thickness were stained with H-E, von Kossa, or TRAP. TUNEL staining was performed using the ApopTag Peroxidase In Situ Apoptosis Detection Kit (Sigma Aldrich, St. Louis, MO). To analyze BrdU incorporation, we subcutaneously injected BrdU into the back of pregnant mice at E16.5 at 100 μg/g body weight 1 h before sacrifice, and detected BrdU incorporation using a BrdU staining kit (Invitrogen, Carlsbad, CA). The sections were counterstained with hematoxylin. For safranin O staining, the sections were stained with hematoxylin, fast green, and safranin O. The area positive for TRAP or safranin O was measured by ImageJ (National Institute of Health). Immunohistochemistry was performed using polyclonal rabbit anti-Col1a1 (Rockland, Limerick, PA), monoclonal

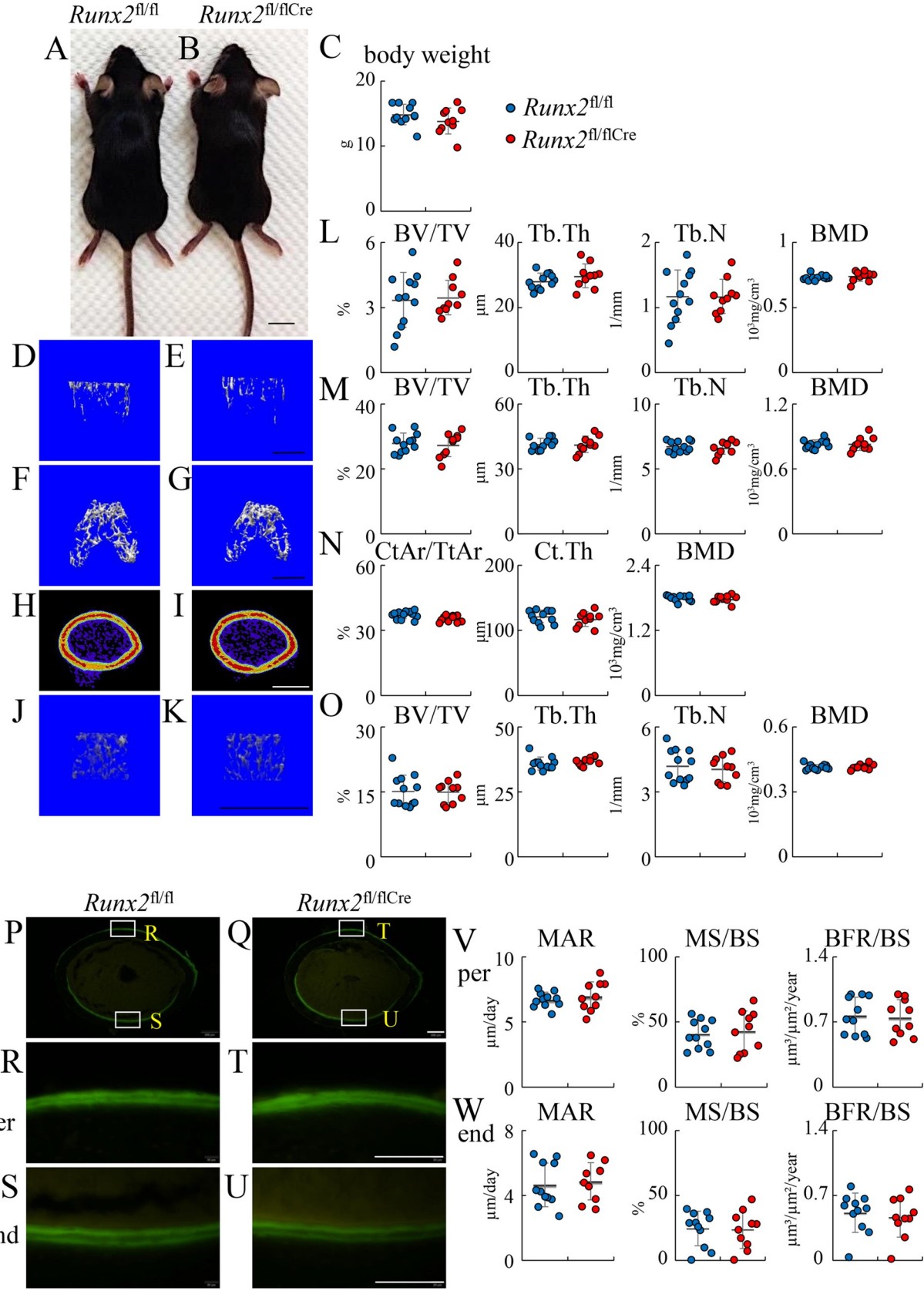

**Fig 10. Micro-CT and bone histomorphometric analyses of *Runx2*<sup>fl/fl</sup> and *Runx2*<sup>fl/flCre</sup> female mice at 6 weeks of age.** (A, B) Physical appearance of *Runx2*<sup>fl/fl</sup> (A) and *Runx2*<sup>fl/flCre</sup> (B) mice. (C) Body weight. (D-K) Three-dimensional images of trabecular bone in the primary ossification center (D, E), trabecular bone in the secondary ossification center (F, G), and cortical bone at mid-diaphysis (H, I) in femurs, and those of trabecular bone in first lumbar vertebrae (J, K) in *Runx2*<sup>fl/fl</sup> (D, F, H, J) and *Runx2*<sup>fl/flCre</sup> (E, G, I, K) mice. The views from the top are shown for the trabecular bone in the secondary ossification center (F, G). (L-O) Trabecular bone volume (BV/TV), trabecular thickness (Tb.Th), trabecular number (Tb.N), and trabecular bone mineral density (BMD) in the primary ossification center (L), secondary ossification center (M), and first lumber vertebrae (O); and cortical area (CtAr/TtAr), cortical thickness (Ct.Th), and cortical bone mineral density (BMD) at mid-diaphysis of femurs (N). (P-W) Bone histomorphometric analyses of cortical bone. Cross-sections from the mid-diaphyses of femurs in *Runx2*<sup>fl/fl</sup> (P, R, S) and *Runx2*<sup>fl/fl Cre</sup> mice (Q, T, U), in which calcein had been injected twice, were analyzed. R and T show periosteum (per) and S and U show endosteum (end). (V, W) Mineral apposition rate (MAR), mineralizing surface (MS/BS), and bone formation rate (BFR/BS) in the periosteum (V) and endosteum (W). BS, bone surface. Scale bars: 1 cm (A, B), 1 mm (D-K), 200 μm (P, Q) and 100 μm (R–U). The number of mice analyzed: *Runx2*<sup>fl/fl</sup>: 12, *Runx2*<sup>fl/fl/Cre</sup>: 10. Data are the mean ± SD.

rabbit anti-Runx2 (Cell Signaling, Danvers, MA), polyclonal rabbit anti-Sp7 (Abcam, Cambridge, UK), and monoclonal mouse anti-VEGF (Santa Cruz, Dallas, TX) antibodies. The secondary antibodies were Histofine Stain MAXPO (R) (Nichirei, Tokyo, Japan) for the former three and Histofine Mouse Stain Kit (Nichirei) for the last. We carried out in situ hybridization using mouse *Col2a1*, *Col10a1*, *Ibsp*, *Mmp13*, *Spp1*, *Col1a1*, and *Bglap2* antisense probes as described previously [2]. The intensities on in situ hybridization were measured by ImageJ. In von Kossa, TUNEL, in situ hybridization, and immunohistochemical analyses, the sections were counterstained with methyl green. For ALP staining, CD34 immunohistochemistry, and the analyses by confocal microscopy, embryos and postnatal mice were euthanized and fixed with 4% paraformaldehyde at 4°C for 2 hours, washed with PBS at 4°C for 1 hour, immersed in 20% sucrose at 4°C overnight, embedded in O. C. T. compound (Sakura Finetek, Tokyo, Japan), frozen in a refrigerated installation (Rikakikai UT-2000F, Tokyo, Japan) containing -100°C hexane and pentane (10:3), and sectioned at 7-μm thickness using Leica CM3050S (Leica Biosystems, Tokyo, Japan). For ALP staining, the cryosections were stained with the solution containing 0.1 mg/ml of Naphthol AS–MX phosphate (Sigma Aldrich), 0.05% N, N–dimethylformamide (Wako, Osaka, Japan), 0.1 M Tris–Hcl (PH: 8.5), and 0.6 mg/ml of Fast BB salt (Sigma Aldrich). The sections were mounted without alcohol treatment after counterstaining with nuclear fast red. The cryosections were also used for immunohistochemistry using anti-CD34 antibody (Hycult Biotech, Uden, Netherlands), and the secondary antibody, Histofine Stain MAXPO (Rat) (Nichirei, Tokyo, Japan). The sections were counterstained with methyl green. The density of blood vessels at the diaphysis of femurs was quantified using Image J software based on CD34 staining. Fluorescent images of tdTomato and mTFP were acquired by confocal microscopy (LSM 800, ZEISS, Oberkochen, Germany). To examine the expression of the LacZ reporter gene, whole mount β-galactosidase staining was performed for *Runx2*<sup>fl/+Cre LacZ</sup> and *Runx2*<sup>fl/flCre LacZ</sup> mice at E15.5, E16.5, and E17.5. Limbs were dissected from embryos and fixed with 4% paraformaldehyde/0.1 M phosphate buffer by immersion at 4°C for 1 h. After washing with PBS 3 times for 1 h, the limbs were incubated in X-Gal solution containing 0.1 M phosphate buffer (pH 7.4), 1 mg/ml of X-gal, 5 mM $K_3Fe(CN)_6$, 5 mM $K_4Fe(CN)_6$, and 2 mM $MgCl_2$ at 37°C overnight with gentle shaking. After staining, the limbs from embryos were decalcified with 10% EDTA for 24 h. After dehydration, the limbs were embedded in paraffin and sectioned at 7-μm thickness. The sections were counterstained with eosin.

## Laser capture microdissection and real-time reverse transcription (RT)-PCR

The hind limbs at E15.5 were separated from embryos and immediately embedded in O. C. T. compound (Sakura Finetek, Tokyo, Japan), and stored at –80°C. The blocks were sectioned at

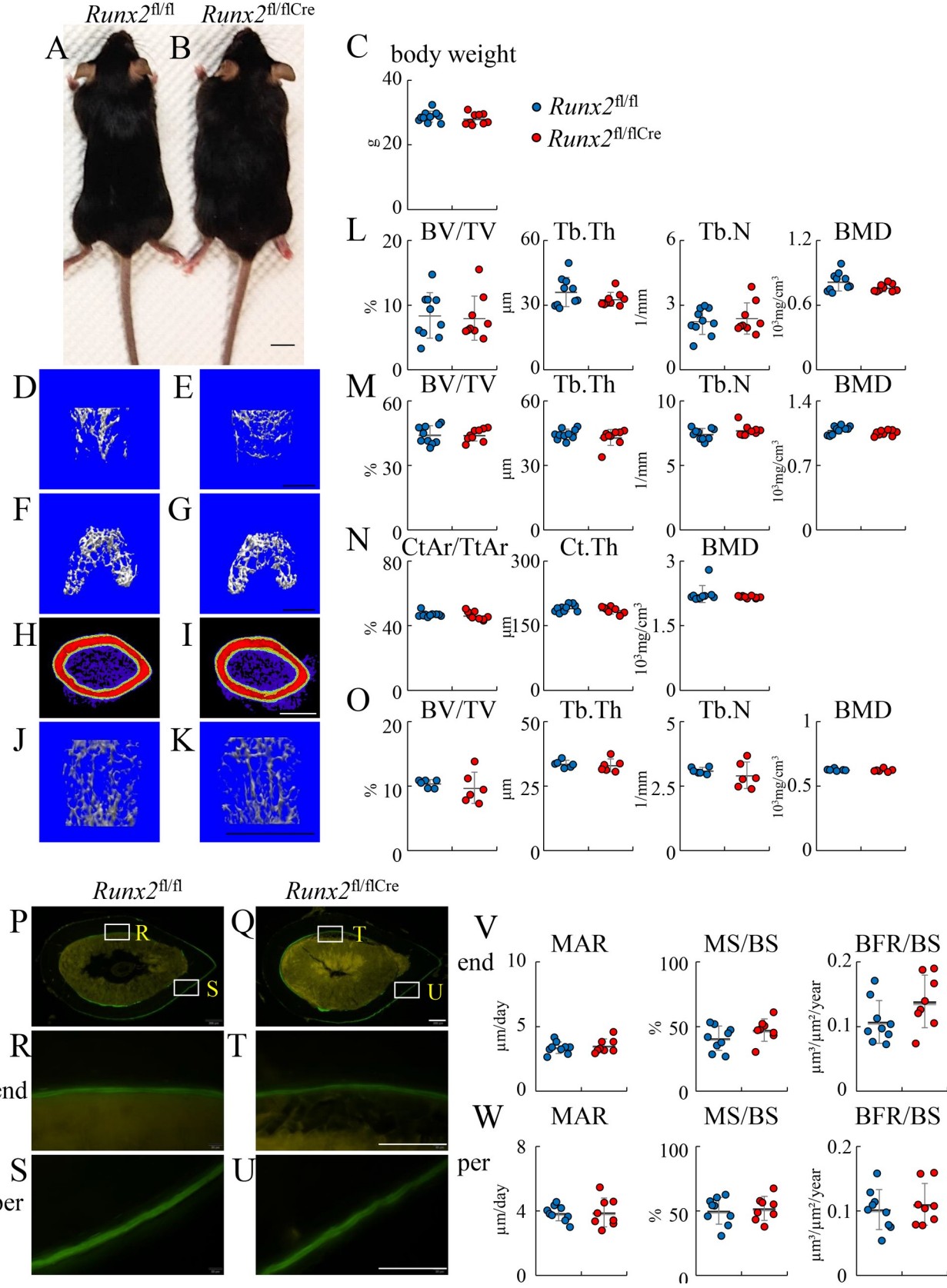

**Fig 11. Micro-CT and bone histomorphometric analyses of *Runx2*<sup>fl/fl</sup> and *Runx2*<sup>fl/flCre</sup> male mice at 20 weeks of age.** (A, B) Physical appearance of *Runx2*<sup>fl/fl</sup> (A) and *Runx2*<sup>fl/flCre</sup> (B) mice. (C) Body weight. (D-K) Three-dimensional images of trabecular bone in the primary ossification center (D, E), trabecular bone in the secondary ossification center (F, G), and cortical bone at mid-diaphysis (H, I) in femurs, and those of trabecular bone in first lumbar vertebrae (J, K) in *Runx2*<sup>fl/fl</sup> (D, F, H, J) and *Runx2*<sup>fl/flCre</sup> (E, G, I, K) mice. The views from the top are shown for the trabecular bone in the secondary ossification center (F, G). (L-O) Trabecular bone volume (BV/TV), trabecular thickness (Tb.Th), trabecular number (Tb.N), and trabecular bone mineral density (BMD) in the primary ossification center (L), secondary ossification center (M), and first lumber vertebrae (O); and cortical area (CtAr/TtAr), cortical thickness (Ct.Th), and cortical bone mineral density (BMD) at mid-diaphysis of femurs (N). (P-W) Bone histomorphometric analyses of cortical bone. Cross-sections from the mid-diaphyses of femurs in *Runx2*<sup>fl/fl</sup> (P, R, S) and *Runx2*<sup>fl/fl Cre</sup> mice (Q, T, U) were analyzed. R and T show endosteum (end) and S and U show periosteum (per). (V, W) Mineral apposition rate (MAR), mineralizing surface (MS/BS), and bone formation rate (BFR/BS) in the endosteum (V) and periosteum (W). BS, bone surface. Scale bars: 1 cm (A, B), 1 mm (D-K), 200 μm (P, Q) and 100 μm (R–U). The number of mice analyzed: *Runx2*<sup>fl/fl</sup>: 10, *Runx2*<sup>fl/fl/Cre</sup>: 9 in femurs and *Runx2*<sup>fl/fl</sup>: 6, *Runx2*<sup>fl/fl/Cre</sup>: 6 in vertebrae. Data are the mean ± SD.

10 μm thickness using Leica CM3050S (Leica, Wetzlar, Germany). The sections were fixed in cold ethanol/acetic acid (19:1) for 3 minutes, washed twice with DDW, then kept in 100% ethanol on ice for laser capture microdissection. Hypertrophic and terminal hypertrophic chondrocyte layers were collected using Leica LMD7 (Leica, Wetzlar, Germany), and RNA was extracted using Arcturus PicoPure RNA Isolation Kit (Thermo Fisher Scientific, Waltham, CM). Real-time RT-PCR was performed using a THUNDERBIRD SYBR qPCR Mix (Toyobo, Osaka, Japan) and Light Cycler 480 q-PCR machine (Roche Diagnostics, Tokyo, Japan). Primer sequences are shown in S1 Table. We normalized the values obtained to those of *Actb*.

## Skeletal and micro-CT analyses

Skeletal preparations were performed as described previously [6]. Quantitative micro-CT analysis was performed using a micro-CT system (R_mCT, Rigaku Corporation, Tokyo, Japan). Data from scanned slices were used for 3-dimensional analysis to calculate morphometric parameters in femurs and first lumbar vertebrae. Femoral trabecular bone parameters were measured on a distal femoral metaphysis, and approximately 2.4 mm (0.5 mm far from the growth plate) was cranio-caudally scanned and 200 slices in 12-μm increments were taken at 6 and 20 weeks of age. Thirty slices were taken in the regions in the secondary calcification center of femurs at 6 and 20 weeks of age, as shown in S5D–S5G Fig. For analysis of the first lumbar vertebrae, 65 and 100 slices were taken at 6 and 20 weeks of age, respectively. For analysis of newborn mice, 260 slices were taken in the whole femurs and 60 slices were taken in the secondary spongiosa. The threshold of the mineral density was 500 mg/cm³ at 6 and 20 weeks of age, and 300 mg/cm³ at the newborn stage.

## Bone histomorphometric analysis

Mice were intraperitoneally injected 20 mg per kg body weight of calcein at either 5 and 2 d or 7 and 2 d before sacrifice in the analysis of 6- or 12-week-old mice, respectively. Mice were euthanized, and the right tibiae and lumbar vertebrae (L3-L5) were harvested and fixed in 70% ethanol for 3 days. Fixed bones were dehydrated with graded ethanol and infiltrated and embedded in the mixture of methylmethacrylate and 2-hydroxyethyl methacrylate (both from Fujifilm Wako pure chemical, Osaka, Japan). Undecalcified 4-μm-thick sections were obtained with a microtome and left unstained for the measurement of fluorochrome labeling. Consecutive sections were stained with either von Kossa method for image capture using cellSens software (Olympus, Tokyo, Japan) or 0.05% toluidine blue (pH7.0) for the measurement of osteoid, osteoblasts and osteoclasts. The bone histomorphometric analysis was performed in proximal tibiae and lumbar vertebra under 200X magnification within a 0.75 mm high X 0.7 mm wide region located 300 μm apart from the growth plate using Histometry RT camera analyzing software (System-supply, Nagano, Japan). 20-μm cross sections from mid-diaphyses of

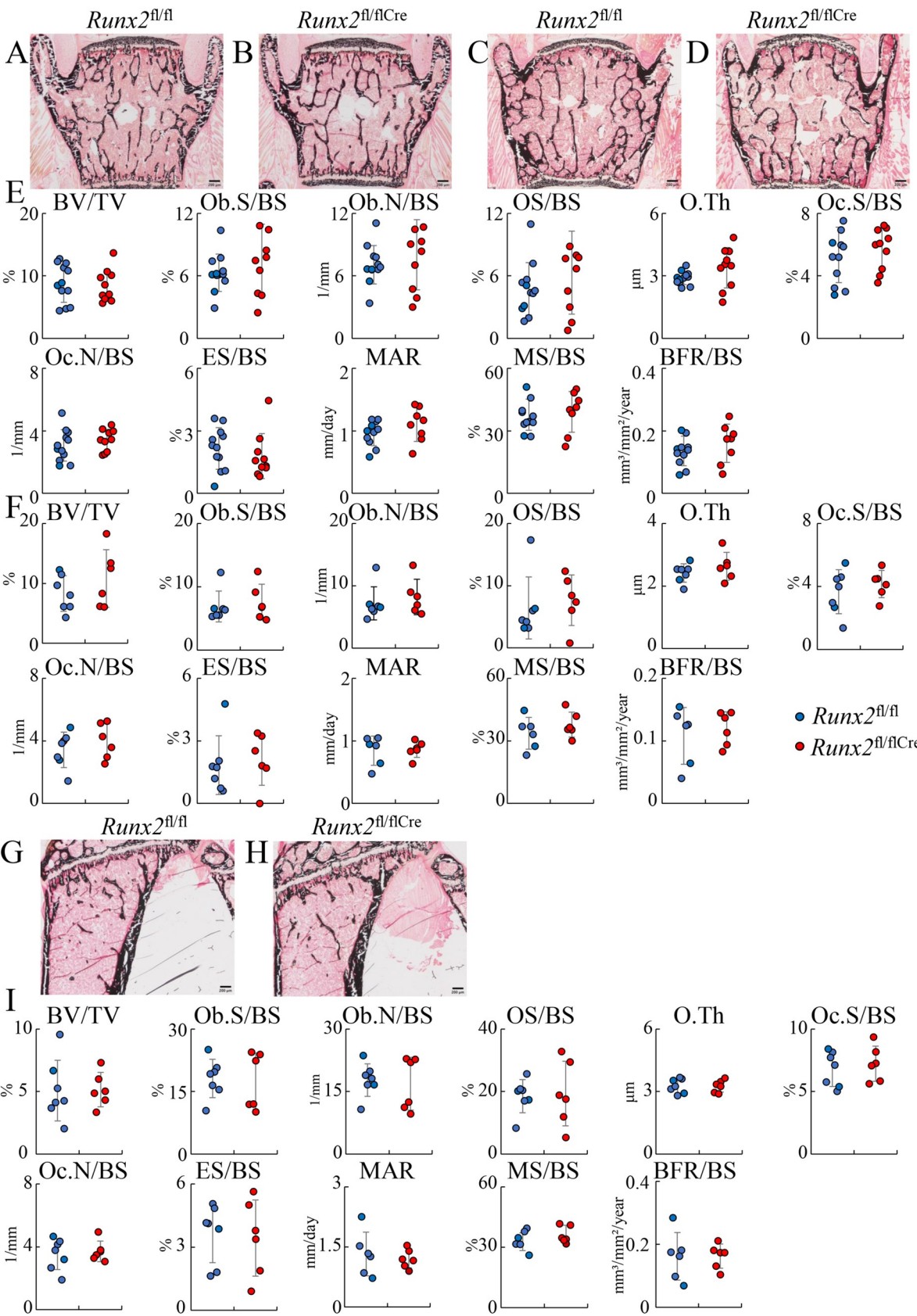

**Fig 12. Bone histomorphometric analyses of vertebrae and tibiae in females at 6 and 12 weeks of age.** (A-D) H-E and von Kossa staining of lumbar vertebra of *Runx2*<sup>fl/fl</sup> (A, C) and *Runx2*<sup>fl/fl Cre</sup> (B, D) mice at 6 weeks (A, B) and 12 weeks (B, D) of age. (E and F) The trabecular bone volume (bone volume/tissue volume, BV/TV), osteoblast surface (Ob.S/BS), osteoblast number (Ob.N/BS), osteoid surface (OS/BS), osteoid thickness (O.Th), osteoclast surface (Oc.S/BS), osteoclast number (Oc.N/BS), eroded surface (ES/BS), mineral apposition rate (MAR), mineralizing surface (MS/BS), bone formation rate (BFR/BS) at 6 weeks (E) and 12 weeks (F) of age. BS, bone surface. (G and H) H-E and von Kossa staining of tibiae in *Runx2*<sup>fl/fl</sup> (G) and *Runx2*<sup>fl/fl Cre</sup> (H) mice at 12 weeks of age. (I) Bone histomorphometric analysis of tibiae. Scale bars: 200 μm. The number of mice analyzed: *Runx2*<sup>fl/fl</sup>: 12, *Runx2*<sup>fl/fl Cre</sup>: 10 at 6 weeks of age; *Runx2*<sup>fl/fl</sup>: 7, *Runx2*<sup>fl/fl Cre</sup>: 6 at 12 weeks of age.

femurs were used for bone histomorphometric analysis of cortical bone. The structural, dynamic, and cellular parameters were calculated and expressed according to the standard nomenclature [36].

## Biomechanical testing

Femurs were isolated from females at 12 weeks of age, wrapped with Kimwipes dipped in saline, and stored in a –30˚C freezer. The three-point bending test was performed in Kureha Special Laboratory (Tokyo, Japan). After thawing the femurs, a load was applied vertically to the midshaft with a constant rate of displacement of 2 mm/min until fracture using MZ-500D (Maruto Testing Machine Co., Tokyo, Japan). A support span of 6 mm was used.

## Statistical analysis

Values are shown as the mean ± SD. Statistical analyses of two groups were performed by the Student's t-test, and those of three groups were performed by analysis of variance and the Tukey-Kramer *post hoc* test, A p-value of less than 0.05 was considered significant.

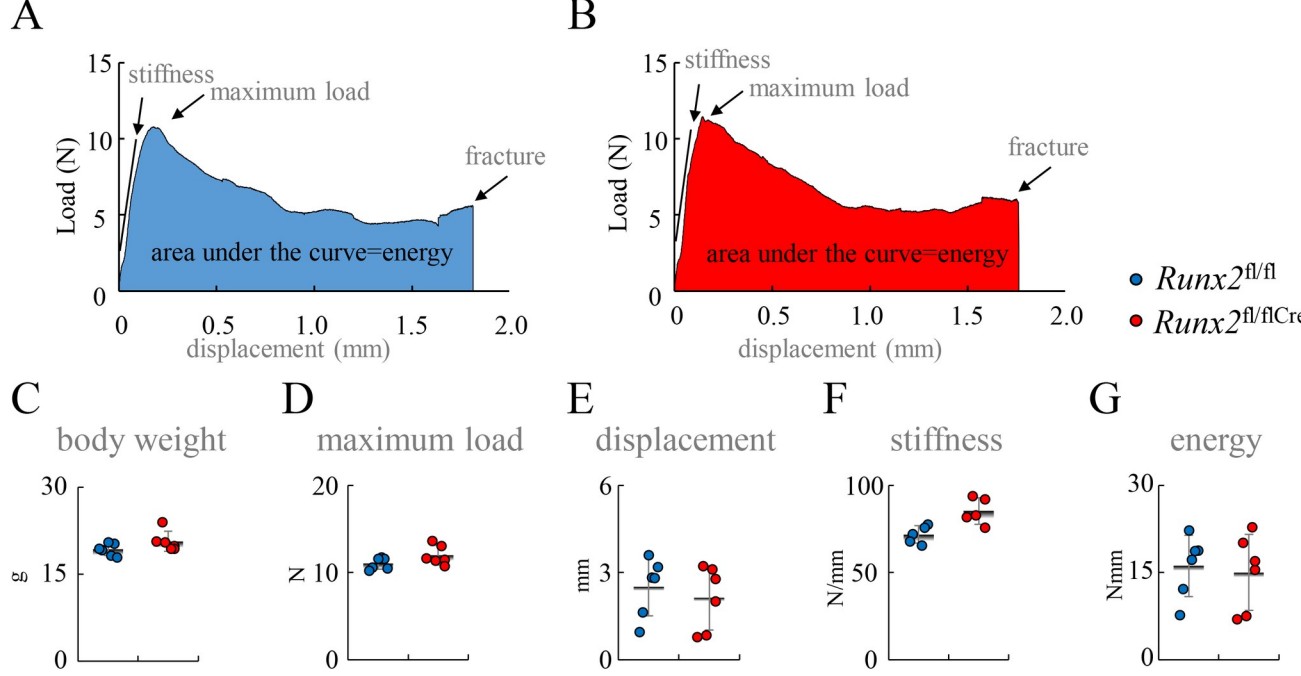

**Fig 13. The three-point bending test.** (A and B) Representative load-displacement curves for female *Runx2*<sup>fl/fl</sup> (A) and *Runx2*<sup>fl/flCre</sup> (B) mice at 12 weeks of age. (C-G) Body weight (C), maximum load (D), displacement (E), stiffness (the slope of the linear part of the load) (F), and energy to failure (shaded area under the curve) (G) in female *Runx2*<sup>fl/fl</sup> and *Runx2*<sup>fl/flCre</sup> mice. The number of mice analyzed: *Runx2*<sup>fl/fl</sup>: 6, *Runx2*<sup>fl/flCre</sup>: 6.

## Supporting information

**S1 Fig. Generation of *Runx2* flox mice.** (A) Targeting vector and strategy for generating a *Runx2*-flox mouse line. (B) Southern blot analyses of genomic DNA from wild-type and targeted ES cells digested with *XbaI* or *SacI* using 5' or 3' probe shown in (A), respectively. (C) PCR for genotyping to detect wild-type and floxed alleles in mice using F and R primers.
(PDF)

**S2 Fig. Immunohistochemistry of Hmgb1.** Immunohistochemistry using anti-Hmgb1 antibody was performed in femoral sections of *Runx2*^fl/fl^ (A, C, E, G) and *Runx2*^fl/fl Cre^ (B, D, F, H) mice at E16.5. (A-D) Staining with DAPI. (E-H) Immunohistochemical analysis using anti-Hmgb1 antibody. The boxed regions in A, B, E, F were magnified in C, D, G, H, respectively. The dotted lines in E and F show the border between the growth plate and bone marrow. The boxed cells in G and H were magnified in the windows. Scale bars: 50 μm (A, B, E, F), 20 μm (C, D, G, H). (I) Cell area/total area, DyLight^+^ area, and intensity of DyLight in the boxed regions without bone marrow area in E and H (hypertrophic and terminal hypertrophic chondrocyte layers) were compared between *Runx2*^fl/fl^ and *Runx2*^fl/fl Cre^ mice. The number of mice analyzed: *Runx2*^fl/fl^: 4, *Runx2*^fl/fl Cre^: 4. Monoclonal rabbit anti-Hmgb1 antibody (Cell Signaling, Danvers, MA) and goat anti-rabbit IgG H&L (DyLight 488) secondary antibody (Abcam, Cambridge, UK) were used for the immunohistochemical analysis.
(PDF)

**S3 Fig. β-galactosidase staining and immunohistochemistry of Col1a1.** (A-C) Negative control for β-galactosidase staining using femoral sections from *Runx2*^fl/fl^ mice at E15.5 (A), *Runx2*^fl/+^ mice at E16.5 (B), and *Runx2*^fl/+ LacZ^ mice at E17.5 (C). The boxed regions in A-C and A'-C' were magnified in A'-C' and A"-C", respectively. (D and E) Immunohistochemical analysis of femoral sections using anti-Col1a1 antibody in *Runx2*^fl/fl^ (D) and *Runx2*^fl/flCre^ (E) embryos at E15.5. Scale bars: 100 μm (A–C', D, E), 50 μm (A"–C"). Two mice for each genotype were analyzed.
(PDF)

**S4 Fig. Generation of 2.3 kb *Co1a1* promoter tdTomato transgenic mice.** (A) Construct of the transgene. * Intron from SV40 containing splice donor and acceptor sites. ** polyadenylation signal from SV40. (B) Whole embryo at E14.5. (C-E) Frozen section of femur at P2. The boxed regions in C were magnified in D and E. Pictures were taken using fluorescence microscope (BZ-X710, KEYENCE, Osaka Japan). Scale bars: 1 mm (B), 500 μm (C), 100 μm (D, E).
(PDF)

**S5 Fig. β-galactosidase staining and micro-CT analysis.** (A-C) β-galactosidase staining of femoral sections from *Runx2*^fl/fl^ (A), *Runx2*^fl/+Cre LacZ^ (B), and *Runx2*^fl/flCre LacZ^ (C) mice at 3 weeks of age. The boxed regions in A-C and A'-C' were magnified in A'-C' and A"-C", respectively. The secondary ossification center is shown. Arrows in B" indicate osteoblastic cells. (D-G) Micro-CT images of femurs at 6 and 20 weeks of age. The trabecular bone between the two lines with the distance of 0.36 mm in the secondary ossification center was analyzed in Figs 10M and 11M. Scale bars: 200 μm (A-C), 20 μm (A'-C'), 10 μm (A"-C"), 0.5 mm (D-G).
(PDF)

**S1 Table. Primer sequences for RT-PCR.**
(PDF)

## Acknowledgments

We thank Dr. I Imayoshi for the permission to obtain Rosa26-CAG-loxP-mTFP1 reporter mice from RIKEN Bioresource Research Center and M. Maki for secretarial assistance.

## Author Contributions

**Data curation:** Xin Qin, Qing Jiang, Toshihisa Komori.

**Formal analysis:** Xin Qin, Qing Jiang, Kenichi Nagano.

**Funding acquisition:** Xin Qin, Qing Jiang, Toshihisa Komori.

**Investigation:** Xin Qin, Kenichi Nagano, Takeshi Moriishi, Toshihiro Miyazaki, Hisato Komori, Chiharu Sakane, Hitomi Kaneko.

**Methodology:** Xin Qin, Qing Jiang, Takeshi Moriishi, Toshihiro Miyazaki, Hisato Komori, Kosei Ito, Klaus von der Mark.

**Software:** Xin Qin, Qing Jiang, Kenichi Nagano, Takeshi Moriishi.

**Supervision:** Toshihisa Komori.

**Validation:** Xin Qin, Qing Jiang, Toshihisa Komori.

**Writing – original draft:** Toshihisa Komori.

**Writing – review & editing:** Toshihisa Komori.

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
