## [Decision Letter · Decision Letter 0]

2 Feb 2020

Dear Dr Komori,

Thank you very much for submitting your Research Article entitled 'Runx2 is required for the transdifferentiation of chondrocytes into osteoblasts, but young mice acquire normal bone mass without transdifferentiation' to PLOS Genetics. Your manuscript was fully evaluated at the editorial level and by independent peer reviewers. The reviewers appreciated the attention to an important problem, but raised some substantial concerns about the current manuscript. Based on the reviews, we will not be able to accept this version of the manuscript, but we would be willing to review again a much-revised version. We cannot, of course, promise publication at that time.

If you decide to revise the manuscript for further consideration at PLOS Genetics, please aim to resubmit within the next 60 days, unless it will take extra time to address the concerns of the reviewers, in which case we would appreciate an expected resubmission date by email to plosgenetics@plos.org.

[LINK]

We are sorry that we cannot be more positive about your manuscript at this stage. Please do not hesitate to contact us if you have any concerns or questions.

Yours sincerely,

Fanxin Long

Guest Editor

PLOS Genetics

Gregory Barsh

Editor-in-Chief

PLOS Genetics

Reviewer's Responses to Questions

**Comments to the Authors:**

Reviewer #1: This paper explores the functions of Runx2 in hypertrophic chondrocytes through conditional inactivation with Col10a1-Cre. This work builds on previous work showing that Runx2 is necessary for chondrogenesis and osteogenesis. In particular Runx2 ko mice only form cartilage, the chondrocytes fail to differentiate into hypertrophic chondrocytes and vascular invasion of the cartilage anlage does not occur. Interestingly in this work, with inactivation of Runx2 in hypertrophic chondrocytes, vascular invasion still occurs, hypertrophy still occurs, but hypertrophic chondrocytes do not differentiate into trabecular osteoblasts, and apoptosis hypertrophic chondrocytes was increased. However, as mice age, the bone phenotypes normalized and no differences between controls and knockouts were observed in adult mice.

Overall, the study is rigorously performed and the conclusions are consistent with the data. The overall knowledge gained from this study is somewhat incremental, in that the main conclusion is that Runx2 is necessary for osteoblast differentiation.

Major Comments

In figure 2, markers of the primary spongiosa, Mmp13, Ibsp, and Spp1, are reduced or absent at E16.5. Is this a delay in formation of the primary spongiosa or does it never form? Analysis of later time points with these markers should address this.

It is shown that Vegfa is reduced in the primary spongiosa but that vascular invasion is normal. However, it is not clear if there is still a delay in the vascular invasion in the Runx2 conditional KO mice. The normal vascular invasion should be demonstrated with an immunostain for endothelial cells.

How does the reduced expression of Vegfa reported here compare with the phenotype of chondrocyte-specific inactivation of Vegfa?

The mechanisms that could explain the increased osteoclasts in Runx2 conditional KO mice should be investigated by examining expression of RANKL, OPG and GM-CSF compared to expression of DAMPs.

The failure of transdifferentiation should be assessed by co-staining for the lineage marker, bGal and an osteoblast markers and quantifying the ratio of lineage marked osteoblasts to total osteoblasts in the primary spongiosa in embryonic and mature bone. An antibody to bGal should exclude staining of osteoclasts. Alternatively, a fluorescent lineage reporter could be used instead of LacZ. The literature suggests that transdifferentiation only accounts for about 20% of osteoblasts in the primary spongiosa. This should be rigorously evaluated here.

microCT analysis shows that long bones and vertebra have normal properties at 6 and 20 weeks of age. It may be worth showing that the strength and material properties of these bones are also normal at these time points.

Minor comments

Skeletal preps in figure 1 should show high magnification of selected skeletal elements.

Reviewer #2: In this study, Qin and colleagues report that conditional loss of Runx2 in hypertrophic chondrocytes does not affect the formation of the bone collar and blood vessel invasion of the developing growth plate but impairs the transdifferentiation of hypertrophic chondrocytes into osteoblasts at the primary spongiosa site. Notably, however, bone mass accrual in both the trabecular and cortical compartments of young adult mice appeared to be normal as shown by microCT analysis.

The paper is interesting, but numerous issues need to be addressed to strengthen the authors’ conclusions and their biological significance.

Specific comments:

1. The Title is confusing and misleading.

2. The Text needs careful editing. In particular, the use of “..under the growth plate..” terminology is unclear. The authors may consider replacing it with “..at the border between the growth plate and primary spongiosa..” or similar sentence.

3. Findings in Runx2fl/+Cre mice need to be shown to exclude the possibility that some aspects of the phenotype could be secondary to the expression of Cre recombinase per se.

4. The authors need to specify what bones have been analyzed throughout the Text.

5. Figure 3 panels Q-V: It is not clear where TUNEL positive cells have been quantified, growth plate, or bone marrow? Please, specify.

6. Figure 4: VEGF IHC is not very convincing. It would be helpful to show the negative controls. In addition, VEGF mRNA should also be investigated to complement the IHC findings. Lastly, the number of blood vessels at the border between the growth plate and primary spongiosa should be quantified.

7. It would be helpful if the authors could comment on the formation of the secondary ossification center and transdifferentiation of hypertrophic chondrocytes into osteoblasts at that site.

8. More importantly, the microCT data need to be completed by static and dynamic histomorphometry analysis to strengthen the authors’ conclusion that bone mass accrual is indeed normal in young mutant mice.

9. In principle, it would be helpful to analyze the bone phenotype in both males and females. At the minimum, the sex of the mice used for analysis should be specified.

Reviewer #3: In this manuscript, authors investigated the role of Runx2 during transdifferentiation of chondrocytes into osteoblast. Conditional KO of Runx2 in hypertrophic chondrocyte impaired the transdifferentiation of chondrocytes, but did not impair the vascular invasion into cartilage. Moreover, authors also indicated the fundamental role of transdifferentiation of chondrocytes during embryonal and adult bone formation. The manuscript is well written and generally easy to comprehend. However, there are several limitations of the study. This for sure could increase the value of the manuscript.

Major comments

1. In this report, there was a change in the extent of VEGF expression, but no change in vascular invasion. However, it is difficult to see that the presented image did not change the invasion of the blood vessels. Vascular endothelium should be stained by such as PECAM1 or quantitatively evaluated. In addition, the reason why there was no difference in vascular invasion should be also discussed.

2. In Runx2fl/fl cre mice, the apoptosis of hypertrophic chondrocyte was enhanced, however apoptosis itself is a phenomenon normally observed in bone differentiation. It is necessary to show or discuss the mechanism. The changes of known factors involved in chondrocyte apoptosis such as Bcl-2 (Oshima et al., JBC 2008.) might be helpful.

3. Osteoclast increased in Runx2fl/fl cre mice. Number of osteoclasts might have strong impact on bone formation and remodeling. The authors discussed that necrosis had increased DAMPs and induced osteoclastgenesis. How did the increase of osteoclast have an effect on bone formation in this model? This issue should be discuss.

Minor comments

1. Also comment or discuss the bones other than the femur in Runx2fl/fl cre mice.

2. In Fig 5, Runx2fl/fl cre and Runx2fl/+ cre were compared. Please show or comment about the differences of the phenotype in Runx2fl/+ cre and Runx2fl/+ or wild type.

3. The numbers in the figures are included in the histological images and are difficult to see. Make it easier to see as much as possible.

**Have all data underlying the figures and results presented in the manuscript been provided?**

Reviewer #1: Yes

Reviewer #2: Yes

Reviewer #3: Yes

PLOS authors have the option to publish the peer review history of their article (what does this mean?). If published, this will include your full peer review and any attached files.

Reviewer #1: No

Reviewer #2: No

Reviewer #3: No

---

## [Decision Letter · Decision Letter 1]

2 Oct 2020

Dear Dr Komori,

We are pleased to inform you that your manuscript entitled "Runx2 is essential for the transdifferentiation of chondrocytes into osteoblasts" has been editorially accepted for publication in PLOS Genetics. Congratulations!

Yours sincerely,

FANXIN LONG

Associate Editor

PLOS Genetics

Gregory Barsh

Editor-in-Chief

PLOS Genetics

Comments from the reviewers (if applicable):

Reviewer's Responses to Questions

**Comments to the Authors:**

Reviewer #2: In this revised manuscript, the authors have satisfactorily addressed reviewers' concerns.

Reviewer #3: The authors revised the manuscript according to the comments by this reviewer. The quality and impact of the presented data are sufficient to merit publication of the manuscript in PLOS Genetics.

**Have all data underlying the figures and results presented in the manuscript been provided?**

Reviewer #2: Yes

Reviewer #3: Yes

PLOS authors have the option to publish the peer review history of their article (what does this mean?). If published, this will include your full peer review and any attached files.

Reviewer #2: No

Reviewer #3: No

**Data Deposition**

http://datadryad.org/submit?journalID=pgenetics&manu=PGENETICS-D-19-02091R1

**Press Queries**

---

## [Editor Report · Acceptance letter]

19 Nov 2020

PGENETICS-D-19-02091R1 

Runx2 is essential for the transdifferentiation of chondrocytes into osteoblasts 

Dear Dr Komori, 

We are pleased to inform you that your manuscript entitled "Runx2 is essential for the transdifferentiation of chondrocytes into osteoblasts" has been formally accepted for publication in PLOS Genetics! Your manuscript is now with our production department and you will be notified of the publication date in due course.

With kind regards,

Matt Lyles

PLOS Genetics

On behalf of:
